# Acute isolation is associated with increased reward seeking and reward learning in human adolescents

Livia Tomova [1,2,3] ✉, Emily Towner [1,4], Kirsten Thomas[1], Lei Zhang [5,6,7], Stefano Palminteri[8,9] & Sarah-Jayne Blakemore [1,10]

Social connection, a basic human need, is vital during adolescence. How a lack of connection impacts adolescent behaviour is unclear. To address this question, we employed experimental short-term isolation with and without access to virtual social interactions (iso total; iso with media; order counterbalanced, both compared to a separate baseline session). Using computational modelling and linear mixed-effects models, we assessed how isolation impacts self-reported loneliness, reward seeking and reward learning in adolescents (N = 40) aged 16–19 years. Self-reported state loneliness increased as a function of duration of isolation. When participants had access to virtual interactions, they self-reported less state loneliness during isolation. Isolation was associated with faster decisions to exert effort for rewards and improved reward learning. These effects were stronger in participants who reported higher state loneliness following isolation. These results demonstrate that, in adolescents, isolation is associated with higher reward responsiveness, a key driver of motivation and decision-making.

Social connection is a basic human need[1,2]. However, we are not always able to obtain the kind of social contact that satisfies our social needs. Reports on chronic loneliness, the subjectively perceived lack of social connection[1], have shown that, across the world, adolescents are one of the age groups most affected by increasing levels of chronic loneliness[3–5] (although there is some heterogeneity in findings[6]). Over the past two decades, longitudinal research has revealed associations between adolescent chronic loneliness and depression[7–12] and drug use[13–16]. However, the direction of causality is unclear, and bi-directional effects between chronic loneliness and mental ill health have been reported[9].

Experimental designs studying how modulations of experiencing a lack of connection relate to changes in outcomes enable a better understanding of causality, but such approaches have so far been mostly limited to animal models of social isolation (but see[17,18] for human isolation experiments in adults). Social isolation is defined as the objective lack of social connection, while loneliness is defined as the subjective feeling of lacking social connection[1]. These constructs are often related (states of social isolation can

lead to feelings of loneliness[19,20]) but can also be dissociated (e.g., people can feel lonely in a crowd)[21]. Past research has also made a distinction between state loneliness (sometimes also labelled transient loneliness) and trait loneliness (sometimes also defined as chronic loneliness). State loneliness refers to the momentary feeling of loneliness, which can fluctuate in daily life and is influenced by different social contexts and company[19,20,22]. Trait loneliness, on the other hand, is defined as the persisting experience of a discrepancy between one's actual and desired social connections and is typically measured via questionnaires (e.g. the University of California, Los Angeles (UCLA) loneliness scale[23]).

Here we were interested in studying how changes in state loneliness induced via social isolation impact subsequent processing of rewards in adolescents. The domain of reward is of particular interest due to its powerful effects on behaviour[24]. There have been persistent efforts to identify the impacts of social isolation on reward processing in adolescents using animal models of isolation. These studies have consistently shown that isolating adolescent animals primarily affects dopaminergic

[1]Department of Psychology, University of Cambridge, Cambridge, UK. [2]Cardiff University Brain Research Imaging Centre (CUBRIC), Cardiff University, Cardiff, UK. [3]Cardiff University Centre for Human Developmental Science (CUCHDS), Cardiff University, Cardiff, UK. [4]Humanities and Social Sciences, California Institute of Technology, Pasadena, California, USA. [5]Centre for Human Brain Health, School of Psychology, University of Birmingham, Birmingham, UK. [6]Institute for Mental Health, School of Psychology, University of Birmingham, Birmingham, UK. [7]Centre for Developmental Science, School of Psychology, University of Birmingham, Birmingham, UK. [8]Departement d'Etudes Cognitive, Ecole Normale Superieure, Université de Paris Sciences et Lettres, Paris, France. [9]Laboratoire de Neurosciences Cognitives et Computationnelles, Institut National de la Santé et de la Recherche Médicale, Paris, France. [10]Institute of Cognitive Neuroscience, University College London, London, UK. ✉e-mail: tomoval@cardiff.ac.uk; lt503@cam.ac.uk

brain reward circuits and alters reward seeking and reward learning (RL)[25,26].

For example, social isolation in adolescence has been shown to increase sensitivity to social rewards[27] and also increase seeking of food or drug rewards in rodents[28]. Conversely, operant access to social interaction has been shown to prevent drug self-administration in rats[29]. Animal research has further shown that social isolation also changes learning about rewards in adolescent animals. Specifically, studies have found that adolescent isolation leads to improved learning about stimulus-reward associations[30,31] but causes perseveration to initially learned associations when reward contingencies change[32-34]. Thus, animal research suggests that while isolation can improve initial learning about rewards, it diminishes the flexibility of changing initially learned responses.

While these results from animal studies have potential implications for the effects of social isolation on reward processing in humans, it is unclear to what extent these findings can be translated to human adolescents.

In the present study, we aimed to specifically investigate how social isolation in human adolescents affects reward responsiveness in the domains of reward seeking and RL.

We hypothesised, first, that (H1) social isolation would lead to changes in self-report measures of affect, specifically: increased loneliness, craving for social contact, state anxiety and negative mood, and decreased positive mood.

Second, we hypothesised that (H2) Social isolation would increase reward seeking as measured by the willingness to expend effort to obtain rewards and that (H2a) this effect would be stronger when rewards are given in a social context.

Third, we hypothesised that (H3) Social isolation would enhance initial reinforcement learning but diminish reversal learning and that (H3a) this effect would be stronger in the social domain.

In addition to reporting high levels of chronic loneliness, young people also report more social media use than other age groups[35]. It has been proposed that there might be a connection between the use of virtual social interactions and chronic loneliness[36-38], while others have proposed that virtual social interactions might be a potential remedy for rising isolation and chronic loneliness[36,39,40]. So far, research has not provided conclusive answers to this question. Here, we sought to study whether access to virtual social interactions while being isolated from real-life social interactions might remediate the effects of social isolation.

We hypothesised that (4) social media use during isolation would remediate the effects of isolation on self-report measures of affect, reward seeking and RL.

These hypotheses were preregistered on: https://osf.io/kbgsv. This study was part of a larger research project, which included different tasks and tested different hypotheses on the effects of isolation. The present manuscript focuses on hypotheses related to the effects of isolation on self-report measures of affect (H1), reward processing in the domains of reward seeking (H2) and RL (H3) and how access to social media during isolation impacts these effects (H4).

## Methods

As preregistered (https://osf.io/kbgsv; preregistration published 21st May 2021), we determined sample size based on pilot data (N = 19; taken from part of the sample in Tomova et al.[17]) from 18–24-year-olds which showed that short-term isolation affected feelings of loneliness (using a self-report loneliness scale ranging from 0–100) after just four hours of isolation with an effect size (Cohen's d) of 0.47. A power analysis (paired t-test) showed that 38 participants are required to detect a medium effect size of d = 0.47 in our outcome measures to achieve a power of .80 (1-beta) at an alpha of .05.

### Participants

We collected data from 42 participants; 2 participants were unable to complete all experimental sessions, leaving 40 complete datasets (mean age = 17.1; std = 0.9; 22 female, 18 male, gender determined via self-report). We did not collect data on race/ethnicity. Participants were recruited

through online advertisements and flyers. All experimental procedures were approved by the Psychology Research Ethics Committee at the University of Cambridge, UK. Participants signed a consent form describing all experimental procedures before participating in the study. Each participant was compensated with up to £127 (minimum payment for each participant was £107) for participating in three sessions.

### Screening questionnaire

Interested individuals completed a screening questionnaire (using Qualtrics and REDCap[41] Software) to assess eligibility for the study. We also collected information on sex and gender of participants via the online questionnaire. Participants were shown a brief explanation of the terms sex and gender and were asked to provide information on both. In our sample all adolescents described their gender as matching their biological sex. Participants were eligible to take part in the study if they were between 16–19 years of age and had no currently diagnosed mental health disorder. Because the baseline visit involved an MRI scan (see preregistration https://osf.io/kbgsv for details), participants were eligible if they reported no permanently implanted metal in their body and no history of brain damage. Furthermore, because data for this study was collected during a COVID-19 pandemic, we also followed exclusion criteria based on a departmental COVID-19 risk assessment. These criteria excluded participants with chronic underlying health conditions (such as asthma), participants who currently felt ill (or had tested positive for COVID-19), and participants who smoked. As we aimed to study the effects of isolation in a sample of adolescents who have frequent and regular social interactions, we also excluded people who: (1) lived alone; (2) reported high feelings of chronic loneliness on the UCLA loneliness scale[23] (we excluded adolescents with scores >50, which is 2 standard deviations above the mean for an adolescent sample[42]); and/or (3) reported substantially smaller social network sizes than previously reported for an adolescent sample[43] measured via two measures: (a) Number of close friends (the original questionnaire asks for the number of people who give social support[44], which we adapted to the number of close friends to simplify the question for our adolescent sample); and (b) Number of social interactions in the past month: counting face-to-face and virtual social interactions that were primarily social (i.e., excluding professional interactions, like talking to a doctor, teacher, or hairdresser). We excluded participants who reported fewer than two close friends and fewer than 10 social interactions in one month (which is ~7 standard deviations below the previously reported means for adolescents for both measures[43]). The exclusion thresholds for the social network size measures were lower than previously reported[17] because data for this study was collected during the COVID-19 pandemic, which had required social distancing and therefore lower levels of social connectedness were expected throughout the population. Data collection started in April 2021 and finished in February 2022. There were social distancing rules in place but no lockdowns or school closures during data collection. Thus, during data collection, young people were back in school full-time and had regular social interactions.

### Experimental procedures

To test our hypotheses, we employed an experimental approach[17] using social isolation of between three to four hours in human adolescents to induce subjective feelings of state loneliness and assess how they are associated with changes in reward seeking and RL. Each participant underwent three experimental sessions, separated by at least 24 h (days between sessions ranged from 2 to 125 (mean = 32.5, median = 27, standard deviation = 27.51) across participants). Figure 1 shows an overview of the experimental procedures.

Each participant first completed a baseline session during which participants underwent the two reward tasks. Following the baseline session, participants were invited to two isolation sessions (order counterbalanced). One session included up to 4 h of total social isolation (iso total) during which participants had no access to any social interactions (real-life or virtual); another session included up to 4 h of social isolation with access to virtual social interactions (iso with media). Each participant was pseudo-

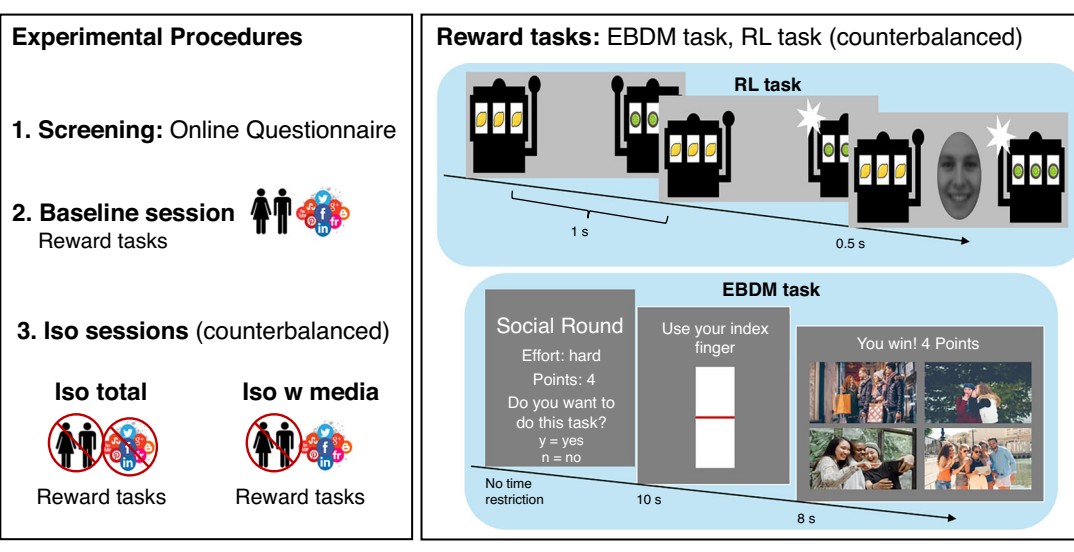

**Fig. 1 | Overview of the experimental procedure. Left panel:** Order of sessions. Individuals first completed an online screening questionnaire to assess eligibility and social connectedness. Eligible participants completed the baseline session, during which they underwent the reward tasks (right panel). Participants were then invited to two isolation sessions (iso total, iso with media; order counterbalanced). At the end of each of the three sessions, participants again carried out the reward tasks. **Right panel:** Timelines of example trials for each reward task. Effort-based decision making (EBDM) task: On each trial, participants were first shown a screen that listed details for the current trial (level of effort (hard, easy), reward (high (4 points), low (1 point)) and context (social, nature)) and were asked whether they wanted to complete the trial. We measured response time (RT) while participants decided if they wanted to complete the trial. If they chose yes, they saw a bar that they were required to pump up in 10 s (the amount of effort required was calibrated to each participant's maximum effort levels assessed at the start of the task in each session) and, if they were successful, they saw a screen indicating how much they had won (1 or 4 points) and depicting images of social interactions or nature landscapes, depending on context (low reward = 1 image; high reward = 4 images). Reward Learning (RL) task: Participants were presented with two slot machines and asked to choose one. After they chose, they received feedback as to whether they had won (indicated by a smiling face or a plus symbol) or did not win (indicated by a neutral face or a zero). Stimulus material displayed in the Figure was taken from a freely available picture database (https://www.pexels.com/).

randomly assigned to one order, with the restriction that the baseline was always first and that each order was approximately equally likely in the full sample. This design allowed us to compare the effects of both types of isolation to a baseline unaffected by the experience of isolation, and to compare the effects of being totally isolated to being isolated with access to virtual interactions while keeping other factors (such as spending time physically alone in a room) constant.

The minimal duration of each isolation session was set to 3 h 30 min and the maximal duration was 4 h. We randomly assigned an added duration of 0–30 min of isolation time to each session (in steps of 5 min, separately for each session), so that participants were not able to predict the precise end of the isolation period in either session. The average isolation duration was similar between sessions (iso total: m = 3 h 46 min, std = 11.2 min; iso with media: m = 3 h 47 min, std = 10.0; t(39) = 0.30, p = 0.765).

During the iso total session, participants had access to a variety of non-social activities (such as games, puzzles, sudoku, books, and drawing/writing supplies). We also encouraged participants to bring activities that they would like to occupy themselves with during the isolation. Examples of what participants chose to bring with them included school homework, a jewellery-making kit, art materials and nail polish.

As we aimed to keep all social interactions during this session to a minimum, participants were given extensive instructions about the isolation procedures and the subsequent behavioural paradigms before starting social isolation. Participants gave their phones and laptops to the experimenter and were guided to a room containing an armchair, a desk, an office chair, and a fridge with a selection of food and beverages. The room had a roof window, which allowed daylight to enter the room but kept participants from seeing other people outside the building. Participants remained in that room for the duration of the isolation and the subsequent behavioural testing. Participants were told that they could spend the isolation time however they wanted, with the restriction that they should fill out the questionnaire every hour and should avoid sleeping during isolation. Participants were reminded to fill out the questionnaires via an alarm sound that went off every hour during isolation and at the end of isolation,

indicating to participants that they should start the behavioural tasks. Participants were provided with a desk computer (with parental controls enabled), allowing them to visit only an online messenger software (www.slack.com) and the webpage containing our online questionnaires. Messaging in Slack was restricted to emergencies (that is, in case participants ran into problems that required assistance from the research team during isolation). During isolation and the subsequent behavioural testing, we monitored participants via a live camera, which allowed a researcher to check in on the participants without interacting with them. The camera only provided a live stream of the participant in the room to the experimenter and did not store any recordings of the participants. Participants were informed about the camera in an information sheet they received before agreeing to participate in the experiment. Participants completed an online questionnaire (see Questionnaires for details) every hour during the social isolation period.

After isolation, participants first filled out questionnaires and then started the behavioural tasks, which they started themselves (i.e., there was no interaction with the experimenter between the end of isolation and the beginning of the behavioural testing). The behavioural testing took place in the same room as the isolation.

After the session, a member of the research team chatted with the participants about their experiences during isolation and made sure participants were not feeling troubled.

Procedures during the iso with media session were the same as in the iso total session, except that participants could bring any electronic device with them into the room and were told they could use them as much as they wanted. Most participants brought their tablet or laptop as well as their phones.

## Questionnaires
**Baseline.** At the beginning of the behavioural testing in the baseline session, participants completed self-report questionnaires assessing trait and state anxiety (the State-Trait Anxiety Index (STAI[45])) and depression (the Center for Epidemiological Studies Depression (CES-D) scale)[46].

Measures of self-reported state loneliness (0–100), social craving (0–100), boredom (0–100) and current affect (using the Positive and Negative Affect Schedule (PANAS[47]) were also acquired at the beginning of the baseline session. In addition, participants were asked whether they drink alcohol and whether they vape (at least once per month). If yes, participants were asked to rate their current urge to drink alcohol and to vape on a scale from (0) not at all to (100) extremely. Following the University COVID-19 risk assessment, participants who reported that they smoke were excluded from participation, and therefore, we did not ask about craving in the domain of smoking.

**Isolation sessions**. Following each isolation session (iso with media and iso total), immediately before participants started the behavioural tasks, we assessed state anxiety and craving for alcohol and vaping. In addition, participants self-reported state loneliness, social craving, boredom and current affect at the beginning of each isolation session (T0) and then after every hour of isolation for 3 h (T1, T2, T3), followed by a final rating at the end of isolation (T4; 3 h 30 min to 4 h of isolation, varying between participants).

After the iso with media session, participants also completed a questionnaire which asked them to report how much they engaged in virtual social interactions during the session (percentage of time spent engaging in virtual social interactions during the session (0–100%)). We also asked participants to indicate which method(s) (texting/messaging, voice calls, video calls, commenting/posting, gaming, other) and platform(s) (Instagram, Facebook, Facebook Messenger, Snapchat, TikTok, Twitter, WhatsApp, other) they mostly used for their virtual social interactions during the session. Participants were asked to list whom they interacted with virtually (i.e., friends, family, acquaintances, romantic partners, and others). Participants could select multiple options and, if they selected "other", they were asked to specify.

### Reward tasks

We used two computerized behavioural tasks to measure reward responsiveness in two domains (i.e., reward seeking and RL).

**Reward seeking**. We used an effort-based decision making (EBDM) task, which was an adapted version of the Effort Expenditure for Rewards Task (EEfRT[48]). In our adapted version, we manipulated and orthogonalized reward and effort. Participants were presented with a series of trials with different combinations of high rewards and low rewards for completing "hard tasks" and "easy tasks". First, participants underwent two calibration trials in which the task measured their maximum effort (the maximum number of button presses performed in 10 s using their index finger). Based on this, "hard task" trials required participants to make 80% of the maximal button presses they managed in 10 s to succeed. "Easy task" trials required participants to make 40% of the maximal button presses they managed in 10 s to succeed. Rewards earned could be either 1 point (low reward) or 4 points (high reward), and participants were presented with combinations of required effort (hard/easy) and potential rewards (high/low) on each trial before deciding whether they wanted to perform that trial's task. Points were converted into monetary value at the end of each session. Each point equalled 2 pence and participants received the cumulative earnings they made across the whole task (which was up to £2.50). Participants were told that each point they won on the task would translate to actual money that they would receive in addition to their participant reimbursement. Participants were not informed about the exact conversion rate but were told that the combined additional wins for both reward tasks in each session could be up to £5. There was no endowment with money in advance as the task did not involve any losses.

In addition to monetary rewards, we added the element of context (nature vs social). This was based on previous research showing that adolescents enjoy looking at pictures depicting social content[49]. The picture stimuli used in the present study were taken from Andrews et al.[49]. At the

beginning of each trial, participants were informed whether that trial was a social or non-social trial (Fig. 1). After successfully performing a trial, participants saw a message indicating how much they won (1 point or 4 points depending on the reward value of that trial) and pictures of landscapes (nature context; e.g., pleasant pictures of mountains, beaches, rivers, etc.) or social pictures (social context; e.g., pleasant pictures of people hugging, laughing together, etc.). Participants either saw 1 picture or 4 pictures, corresponding to the number of points they earned in that trial (i.e., 1 picture for 1 point and 4 pictures for 4 points). This allowed us to measure whether isolation selectively enhanced reward seeking in social vs non-social contexts. In total, there were eight conditions: high effort–high reward; low effort–high reward; high effort–low reward; low effort–low reward, each in either a nature or social context, with six trials per condition (48 trials in total), which were presented in random order.

Participants had to indicate whether or not they wanted to play each trial. There was no time restriction for participants' decisions. If participants completed the trial successfully (i.e., pumped up the bar within the time limit), they saw the number of points they won and the social or nature pictures described above. Pictures were presented for 8 s together with the number of points participants won. After each trial, a fixation cross was presented for 0.5 s.

**Reward learning**. The ability to learn stimulus–reinforcement associations and to reverse them based on probabilistic feedback was measured using a probabilistic reinforcement and reversal learning task[50,51]. In the present task, participants were shown two slot machines and asked to choose between them to obtain a reward. One slot machine was rewarded 80% of the time; the other was rewarded 20% of the time. Participants needed to learn from feedback through trial and error, which slot machine was rewarded more often. After seven trials, the reward contingencies switched, and participants needed to learn the new reward contingencies.

Participants were not informed about the structure of the learning task (i.e., that there would be reversals) or the probabilities of reward. Participants were only told that their task was to find out, by trial and error, which is the more advantageous slot machine and win as many points as possible across the whole task.

Feedback was given via symbols (non-social feedback) and facial expressions (social feedback) in two counterbalanced blocks (28 trials per block, in total 56 trials). Rewards (positive feedback) were represented by either a plus symbol (+; non-social) or a smiling face (social), while the absence of a reward (negative feedback) was represented by a zero symbol (0; non-social) or a neutral face (social).

The two facial pictures (smiling and neutral) were generated by averaging facial pictures of Caucasian young adults. To do this, we used happy and neutral faces from the Averaged Karolinska Directed Emotional Faces set[52]. We averaged the female and male faces for each emotion (happy and neutral) rendering them ambiguous as to gender. Photoshop was used to create the averages. We cropped the images to remove the background/hair and display the face only. Facial pictures were displayed in black and white to match the non-social feedback. Participants were given 1 s to respond on each trial and then received feedback for 0.5 s. A fixation cross was presented for 0.5 s between each trial.

*Counterbalancing*. There were three pairs of stimuli used—the 'fruits' within the slot machines. Pair 1: Cherries & Grapes Pair 2: Watermelon & Banana Pair 3: Lemon & Kiwi. Participants saw a different pair in each of their sessions and the order of which pair participants saw at which session was counterbalanced across participants. Whether or not the first 'correct' response was on the left or the right side was randomized within each session and whether the social or nonsocial feedback came first was also randomized within each session.

Participants also underwent three other tasks during the experiment (a go-no-go task, a peer influence task and a threat learning task (see full description of experimental procedures here: https://osf.io/kbgsv). The

order of the tasks was counterbalanced between participants with the exception that the threat learning task was always presented after the reward tasks reported here so that there would be no effect of threat exposure on reward processing. Results from the other behavioural tasks will be reported elsewhere.

Data collection and analysis were not performed blind to the conditions of the experiments.

### Data analysis

**Self-report measures**. All statistical analyses using self-report questionnaires were conducted on measures taken after 3 h (T3) of isolation. This allowed us to capture participants' affective state after a substantial period of isolation but before they knew it was over. We also ran sensitivity analyses testing if effects differ when using the final time point (T4) which they did not (reported in Supplementary Note 1).

We preregistered to compare effects between all three sessions (baseline, iso total and iso with media) and for this analysis (comparing measures taken at T3 for the two isolation sessions to the baseline measure) we included session (baseline, iso total and iso with media) as a predictor in a mixed effects model to test for differences between sessions on loneliness, social craving, state anxiety, positive mood and negative mood with subject included as random effect. We report the results of this analysis in the SI (Supplementary Note 2).

We also preregistered analysing the effects of isolation on substance craving, but because only 55% of our participants ($N = 22$) reported that they drank alcohol at all, and only two participants reported that they vaped, there was not sufficient power to assess effects of isolation on alcohol or vape craving and we did not perform any analyses on these data.

To test if self-report measures changed throughout the isolation sessions, we employed the following exploratory analyses: first, we compared measures taken at T0 to those taken at T3 using paired sample t-tests.

We chose to compare measures between T0 and T3 (and T4 in Supplementary Note 1) as we wanted to assess whether loneliness (and other self-report measures of affect) were significantly different at the end of isolation (compared to the beginning) shortly before participants underwent the reward tasks.

Second, to assess if the change over time (i.e., the slope) was different between the two isolation sessions, we used linear mixed-effects models with the following predictors: session (iso total and iso with media) and duration (hours of isolation: 0, 1, 2, 3) with subject included as a random effect (allowing intercepts and slopes to vary between participants[53]). Note that we did not include the baseline session in this comparison as this session did not have the relevant measures (i.e., repeated collection of state measures throughout isolation).

The command was: fitlme (Data,'self-report measure ~ (session* duration)+ (session*duration|subjectID)').

We tested whether the data met the assumptions for a linear mixed-effects model by plotting residuals for this and all subsequent analyses.

**Reward Seeking**. Following our preregistration (https://osf.io/zckyn), we calculated the sum of number of played trials and mean response times (RTs) across all trials for the following combinations in each condition (nature or social): high effort–high reward; low effort–high reward; high effort–low reward; low effort–low reward (eight conditions in total). We assessed whether RTs for deciding to play differed between sessions (baseline, iso_total, and iso_media). We also generated means across the different conditions (nature and social) and tested for differences between sessions for each combination (across both nature and social): high effort - high reward; low effort – high reward; high effort–low reward; low effort-low reward. We used mixed effects models to test for differences between sessions to estimate the fixed effects of effort (high, low), reward (high, low) and session (baseline, iso_total, and iso_media) in each context (nature, social) on RTs. Again, with subject included as a random effect (allowing intercepts and slopes to vary between participants[53]). Iso total was the reference session in the model.

Calibration (i.e., the number of button presses participants managed at the beginning of each session from which hard and easy tasks were calculated) was added as a control variable in the model.

The command was: fitlme(Data,'RT ~ (session*effort*reward* context)+(Calibration)+(session*effort*reward*context|subjectID)').

In our preregistration, we also planned to run the same analysis on the number of trials played for each condition. However, we were not able to run this analysis due to the low variance in participants' choice data. More specifically, many participants chose to play almost every trial in the task (73% of all trials were played; between participant standard deviation = 20.2%; 15 participants played 100% of trials in at least one of the sessions). This effect was similar across sessions: i.e., from a total of 28 sessions in which participants played every trial, ten were from baseline sessions, nine from iso total sessions and nine from iso with media sessions. Data with zero variance holds no information, and one session with zero variance effectively compromised our ability to analyse data for that participant (as we would not be able to compare effects of the other sessions to that session). We therefore decided not to analyse choice data and instead focus assessments on RTs when deciding to play a trial (which are indicative of the strength of preference[54]) as the main measure of reward seeking.

A second deviation from our preregistration was that we log-transformed RTs for the mixed effects model analysis in the EBDM task, as prompted by a reviewer, following recommendations from Gelman & Hill 2006[55].

A detailed description of our preregistered analyses and where we deviated from them is provided in Supplementary Note 7).

**Exploratory analyses: Isolation-induced changes in state loneliness and changes in reward seeking**. To further interrogate the driving factors underlying the changes in RTs between the two isolation sessions, we explored whether the difference in self-reported state loneliness between the two isolation sessions (i.e., the difference between iso total and iso with media in the state loneliness ratings after 3 h of isolation) correlated with the difference in reward seeking. We focused on high reward- high effort trials in this analysis as we consider these the most relevant for studying motivation and reward seeking (i.e., capturing participants' willingness to expend effort for rewards; see[48] for a similar rationale). We calculated a Pearson correlation to assess if the change scores between loneliness were correlated with a change in social reward seeking (RTs in high reward, high effort trials, assessed separately for each context (social, nonsocial; Bonferroni corrected $p < 0.025$ (0.05/2)). For this analysis, we used the original (non-log-transformed) data because recommendations from Gelman & Hill 2006[55] are for linear models while here we simply calculated difference scores between sessions and tested if they were correlated. The variables used for the correlations were normally distributed as tested using Shapiro-Wilk tests.

**Reward learning**. Following our preregistration (https://osf.io/zckyn), participants' choices from the reinforcement/reversal learning task were analysed using a computational reinforcement learning and decision-making model for probabilistic reversal learning tasks (Ahn 2017[50], Metha 2020[51]). We deviated from our preregistered analysis plan as we found that our preregistered model did not fit the data well. We therefore implemented a model comparison and selected a model that showed a better fit to the data. We also deviated from our plan by removing the preregistered linear mixed-effects model analysis on participant-level model parameters as we found out that such two-step approaches are not correct (i.e., they show a bias towards the alternative hypothesis[56]).

**Exploratory analyses: behavioural performance**. We assessed whether there was an effect of session on performance during RL. To do that, we measured accuracy during reversal learning, which was quantified as the proportion of correct responses during the task. Because our version of the task included a time limit for responding (1 s), hence making fast responding advantageous, we also assessed whether there was an effect of session on RTs for making choices during the task. We log-transformed

RTs before including them in the linear mixed model analysis following the recommendations of Gelman & Hill 2006[55].

We tested the effects of session (baseline, iso total, iso with media), feedback type (social, non-social) and phase (1, 2, 3, 4; added as a categorical predictor) on accuracy and RTs using mixed effects models. Subject was included as a random effect (allowing intercepts and slopes to vary between participants[53]). Iso total was the reference session in the model.

The command was: fitlme(Data,'outcome ~ (session*feedback type * phase) + (session*feedback type + phase |subjectID)').

We then analysed perseverative errors from each reversal phase (phases 2–4). Perseverative errors were quantified as the proportion of errors choosing the previously reinforced stimulus following each reversal. In each phase, for phases 2–4 (excluding phase 1 because no rule had been learned yet), we assessed the percentage of perseverative errors participants made (sum of errors following reversal/number of trials). We analysed the phases separately as this allowed us to assess the effects of session on updating of previously learned reward contingencies when participants had not experienced any previous reversal of contingencies yet (phase 2) when contingencies reversed back to be the same as during initial acquisition (phase 3) and when participants had to reverse contingencies a second time (phase 4). We ran separate analyses for each phase and report results as significant at Bonferroni corrected $p < 0.017$ (0.05/3).

**Computational modelling.** Following the analysis of performance data, participants' choices from the RL task were analysed using a computational reinforcement learning and decision-making model for probabilistic reversal learning tasks[50].

To assess learning during the task, we first extracted the trial-by-trial responses for each participant and then employed a computational reinforcement learning model for probabilistic reversal learning using a hierarchical Bayesian modelling approach[50] to estimate the learning rates for each participant.

For the group level means and standard deviations we used normal distributions; see Ahn et al.[50] for more details on model specification and parameter declaration[50].

We performed model comparison to select a model that best captured the behaviour of participants. To do that, we first compared model fits between four different reversal learning models on the data from each session. Each model employs the Rescorla-Wagner value update rule[57] but differs in terms of how information is integrated. The Rescorla-Wagner update rule assumes that individuals assign and update internal stimulus value signals based on the prediction error, i.e., the mismatch between outcome (received reward/punishment following choice of this stimulus) and prediction (expected value of choosing this stimulus). The following models were compared:

(i) A reward-punishment model[58], which expands the classic Rescorla-Wagner model of conditioning[57] with separate learning rates for reward and punishment trials, here treating non-wins as punishments:

$$V_{c,t} = \begin{cases} V_{c,t-1} + \eta^{rew}(O_{t-1} - V_{c,t-1}), \text{if} & O_{t-1} > 0 \\ V_{c,t-1} + \eta^{nrew}(O_{t-1} - V_{c,t-1}), \text{if} & O_{t-1} < 0 \end{cases} \quad (1)$$

Where $\eta^{rew}$ is the learning rate for rewards and $\eta^{nrew}$ is the learning rate for non-rewards; O is the outcome received. In this model, only the chosen stimulus value is updated. $V_{c,t}$ is the value of choice c on trial t. $O > 0$ indicates a win and $O < 0$ indicates no win.

(ii) An experience-weighted attraction model[59], which was the model we originally preregistered. This model includes an "experience" weight parameter that balances between past experience and new information to increasingly tip in favour of past experience:

$$n_{c,t} = n_{c,t-1} \times \rho + 1 \quad (2)$$

$$V_{c,t} = (V_{c,t-1} \times \varphi \times n_{c,t-1} + O_{t-1})/n_{c,t} \quad (3)$$

Where, $n_{c,t}$ is the "experience weight" of the chosen stimulus on trial t, which is updated on every trial using the experience decay factor $\rho$. $V_{c,t}$ is the value of choice c on trial t for outcome O received in response to that choice, and $\varphi$ is the decay factor for the previous payoffs. In this model, $\varphi$ is equivalent to the inverse of the learning rate in Rescorla-Wagner models.

(iii) The experience-weighted attraction model with separate learning rates for positive and negative prediction errors.

$$V_{c,t} = \begin{cases} (V_{c,t-1} \times \varphi^{rew} \times n_{c,t-1} + O_{t-1})/n_{c,t}, \text{if} & O_{t-1} > 0 \\ (V_{c,t-1} \times \varphi^{nrew} \times n_{c,t-1} + O_{t-1})/n_{c,t}, \text{if} & O_{t-1} < 0 \end{cases} \quad (4)$$

(iv) A fictitious update (FU) model[60], which includes an update rule for the unchosen option considering the knowledge individuals gain about the unchosen option, here with separate learning rates for positive and negative prediction errors.

$$V_{c,t} = \begin{cases} V_{c,t-1} + \eta^{rew}(O_{t-1} - V_{c,t-1}), \text{if} & O_{t-1} > 0 \\ V_{c,t-1} + \eta^{nrew}(O_{t-1} - V_{c,t-1}), \text{if} & O_{t-1} < 0 \end{cases} \quad (5)$$

$$V_{nc,t} = \begin{cases} V_{nc,t-1} + \eta^{nrew}(-O_{t-1} - V_{nc,t-1}), \text{if} & O_{t-1} > 0 \\ V_{nc,t-1} + \eta^{rew}(-O_{t-1} - V_{nc,t-1}), \text{if} & O_{t-1} < 0 \end{cases} \quad (6)$$

Where the value V of both the chosen c and unchosen nc stimulus is updated with the actual prediction error and the counterfactual prediction error per trial t, respectively. O is the outcome received. The learning rate $\eta$, which is divided into $\eta^{rew}$ (learning rate for rewards; learning_pos) and $\eta^{nrew}$ (learning rate for non-rewards; learning_neg) evidences the magnitude of the value update affected by both positive and negative prediction errors.

Model fitting. For all models, a softmax choice function was used to compute the action probability given the action values. On each trial $t$, the action probability of choosing option A (over B) was defined as follows:

$$p(A) = \frac{1}{1 + e^{-\beta(V_A - V_B)}} \quad (7)$$

Where $\beta$ is the inverse temperature parameter that governs the stochasticity of the choice, computed using inverse logit transfer. Higher $\beta$ values denote decisions driven by relative value, whereas lower $\beta$ values denote more choice stochasticity.

Parameter estimation was performed with hierarchical Bayesian analysis using Stan language in R via modified scripts from the hBayesDM package[50]. Posterior inference was performed using Markov Chain Monte Carlo (MCMC) sampling using the mode of the posterior distribution as the summary parameter for individual participants. Four MCMC chains were used, with 1000 post-warmup iterations per chain, resulting in 4000 valid MCMC posterior samples. Model convergence was assessed by examining R-hat values, an index of the convergence of the chains[61]. R-hat values of all models were lower than 1.05, suggesting MCMC samples were well mixed and converged to stationary distributions.

The models were fitted on all data combined from each session (baseline, iso total and iso with media) and feedback (social, non-social). To take full advantage of the hierarchical Bayesian approach, we fit models to the overall data across all conditions rather than to separate data from each condition to better capture the within-subject design structure and to avoid overestimation of differences among conditions. Specifically, we constructed a design matrix of effect coding to reflect differences among

sessions and conditions:

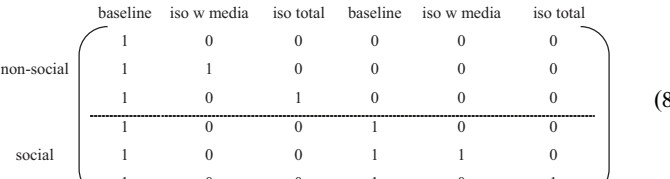

$$(8)$$

Here, the first three rows depicted non-social feedback, and the last three rows depicted social feedback; columns were used to describe the difference between sessions with respect to the baseline session. For example, the second column indicated the difference between iso_media and baseline for non-social feedback. As a result, row summaries indicated the actual parameters for the combinations between sessions and feedback. For example, the second row indicated the parameter in the iso_media condition (e.g., $X_{iso\_media} = X_{baseline} + X_{iso\_media\_diff}$). All other rows and columns followed the same rule. This design matrix was applied to all model parameters in each candidate model under a single overall hierarchical fitting framework.

Model comparison. Comparison of model fit was assessed using Bayesian bootstrap and model averaging, whereby log-likelihoods for each model were evaluated at the posterior simulations and weight was obtained for each model. We compared Bayesian model weights between models (higher weights denoting better model fit) using two model weight metrics: Bayesian stacking of means and Pseudo Bayesian Model Averaging[62]. We found that model fit was best for the FU model (Supplementary Table 2).

Following model selection, we compared the posterior distributions of the group level parameters (learning_pos, learning_neg and beta) for each of the six cases (i.e., the combination of session (baseline, iso total and iso with media) and feedback (social, non-social): baseline-social, baseline-non-social, iso total-social, iso total-nonsocial, iso with media-social and iso with media-nonsocial). We use the Savage-Dickey method[56] to perform hierarchical Bayesian hypothesis testing, which generates a Bayes factor (BF) for each pairwise comparison. The BF is obtained by comparing each difference distribution to a prior distribution that expresses the expectation about the effect size of the pairwise difference $\delta = (\mu2 - \mu1)/\tau$, where $\mu1,2$ are the means of the conditions to be compared and $\tau$ is the group level standard deviation. The null hypothesis specifies that $\delta = 0$, indicating no difference between the conditions. The real difference distributions are then compared to this prior distribution for each case. Specifically, because the null hypothesis is $\delta = 0$, the BF is described as the prior density at $\delta = 0$ divided by the posterior density at $\delta = 0$. Thus, the BF for each pairwise comparison expresses the likelihood of the observed data occurring under the alternative hypothesis (i.e., that the difference between cases is non-zero; labelled as $BF_{10}$). A $BF_{10}$ of 1 represents equal support for the null and alternative hypothesis (inconclusive), while a $BF_{10} < 3$ is weak, $BF_{10} > 3$ is moderate and $BF_{10} > 10$ is strong evidence for the alternative hypothesis[63,64]. We computed the Highest Density Interval (HDI) to calculate the 95% credible intervals for the difference distributions.

Simulations. Whether a certain difference in parameters indicates "better" or "worse" behaviour can be heavily dependent on the task design and the values of the other relevant model parameters[65]. We therefore used data simulations to identify the optimal learning parameters for the present task environment to help with the interpretation of the results. The parameter combinations were taken from a grid spanned by the learning rate ($\eta$; minimum value/maximum value/steps = 0/1/60) and "inverse temperature $\beta$" (0/20/60). Each of these virtual participants then completed 28 trials corresponding to the actual number of trials in our task for each feedback (social, non-social) in each session (baseline, iso total, iso with media) with a reward schedule of 80:20. For each virtual participant, we then calculated the percentage of correct choices as the percentage of choices for the option that

was associated with a higher probability for a reward. To reduce random noise due to the finite number of samples, we smoothed the resulting images with a Gaussian filter (std = 2). We identified the optimal learning rate and inverse temperature as the values that resulted in the highest choice accuracy (average of ten highest accuracy scores in the simulations; Supplementary Fig. 1).

Posterior predictive checks. We simulated data from the FU model and assessed how well it represented real participants' choice data by comparing the two on the single-subject and group mean level. Supplementary Fig. 2A, B shows the results from the posterior predictive checks.

Parameter recovery. We also ran parameter recovery to assess whether the fitting procedure is giving meaningful and identifiable parameter values[66,67]. Posterior group-level mean and standard deviation were extracted as the "ground truth" for each parameter from their posterior distributions and then used for generating $n = 40$ individual level parameters to simulate data. Simulated data was generated from the FU model and the model was then fit to the simulated data. The posterior group-level distribution from each recovered parameter was compared to the ground truth to evaluate parameter recovery[68]. Parameters are considered well recovered if the ground truth is within the 95% HDI of the posterior distribution[69], which was the case for all parameters. Thus, parameter recovery was overall successful for all parameters (see Supplementary Fig. 2C, E for the recovered parameter posterior distributions compared against the ground truth for each parameter).

**Exploratory analyses: Isolation-induced changes in state loneliness and changes in RL.** Analogous to the EBDM task, to further interrogate the driving factors underlying the changes in RL between sessions, we explored whether the difference in self-reported state loneliness between the two isolation sessions (i.e., the difference between iso total and iso with media in the state loneliness ratings after 3 h of isolation) correlated with a difference in RL. We calculated Pearson correlations for each learning rate (learning_pos and learning_neg). For each parameter, we calculated change scores between the iso total and iso with media session and then calculated Pearson correlations for each feedback (social, non-social; Bonferroni corrected $p < 0.025$ (0.05/2)) to assess if the change scores between loneliness and RL were correlated. We used individual-level parameters (learning_pos, learning_neg) for this analysis as past work has shown that correlation analyses are not affected by the two-step approach bias[70]. The variables used for the correlations were normally distributed as tested using Shapiro-Wilk tests.

Data processing, analysis and visualisation were conducted in Python (version: 3.7) using Jupyter Notebook (packages SciPy, Pandas, NumPy and Seaborn). Mixed-effects models were implemented in Matlab 2020a and computational modelling was implemented in RStudio.

## Results
All participants ($N = 40$; age range = 16–19 years; mean age = 17.1; SD = 0.9; 22 female) were socially well-connected in that they reported having frequent social interactions (number of face-to-face or virtual interactions that were primarily social in the past month: mean = 36.0 (SD = 32.2); minimum = 10) and several close relationships (mean = 7.6 (SD = 4.2); minimum = 2). Participants reported average levels of pre-existing chronic loneliness for their age group[23,71] (UCLA loneliness scale: mean = 35.6 (SD = 6.2); maximum=48 out of 80).

During the iso with media session, participants self-reported that, on average, they spent 47% of their time engaging in virtual social interactions during the iso with media session (range 5–100%; mean = 47.35, median = 40, std = 28.3). Most participants (35 out of 40) engaged with virtual social interactions more than 20% of the iso with media session and 18 out of 40 participants (45%) spent more than 50% of the session engaging in virtual social interactions. We asked participants what they used for connecting with others (multiple selections possible) and the majority of participants

reported using instant messaging (37 out of 40). Some participants reported posting (9) and a small number reported engaging in voice calls (3), video calls (2) or gaming (3). Participants mainly used Snapchat (28), Instagram (27) and WhatsApp (23) to connect with others, followed by TikTok (10), Twitter (3), Discord (3) and Facebook/Facebook Messenger (2). The majority of participants connected virtually with friends (38), followed by family (19), romantic partners (13) and acquaintances (4).

### Adolescents show increased state loneliness, boredom and decreased positive mood following isolation

Our analysis testing effects of isolation on self-report measures of affect (H1), showed that after three hours in both isolation sessions, compared with at the start of the session, participants reported significantly increased state loneliness, boredom and decreased positive mood (Table 1; Fig. 2).

Our analysis testing whether social media use during isolation would remediate effects of isolation on self-report measures of affect (H4), showed that the increase in state loneliness and boredom over time was higher in the iso total session than in the iso with media session (state loneliness: beta (b)= $-3.76$, t(302) = $-2.89$, 95% confidence interval (CI) = $-6.33$, $-1.20$, $P = 0.004$; boredom: $b = -6.01$, t(302) = $-3.43$, CI = $-9.45$, $-2.57$, $P < 0.001$). However, the decrease in positive mood over time was not significantly different between the iso with media and iso total session ($b = -0.05$, t(302) = $-0.18$, CI = $-0.67$, 0.57, $P = 0.857$). There was no difference between sessions in measures over time for negative mood ($b = 0.003$, t(302) = 0.02, CI = $-0.25$, 0.26, $P = 0.985$) and social craving ($b = 1.22$, t(302) = 0.92, CI = $-1.38$, 3.81, $P = 0.358$). State anxiety did not differ between sessions (Fig. 2 and see Supplementary Note 2 for details). We report the direct comparison between sessions for the other measures (comparing measures after 3 h of isolation) in the Supplementary Note 2.

### Adolescents show increased reward seeking following isolation

Our analysis testing effects of isolation on reward seeking (H2) and whether social media use during isolation would remediate effects of isolation on reward seeking (H4), showed that there was a main effect of session showing lower RTs in the iso total session compared to baseline ($b = 0.32$, t(732) = 5.06, CI = 0.19,0.44, $P < 0.001$) and compared to the iso with media session ($b = 0.16$, t(732) = 2.55, CI = 0.04,0.28, $P = 0.011$; Fig. 3A). There was also a session * reward * context interaction showing lower RTs in the iso total compared to iso with media session for high reward trials in the social context ($b = -0.26$, t(732) = $-3.0$, CI = 0.09,0.43, $P = 0.003$; Fig. 3B). See Supplementary Table 1 for full results. To help interpret the 3-way interaction, we performed a pairwise comparison directly testing RTs in the high reward condition within the social context, between the iso total and the iso with media session. This showed that RTs in the high reward condition in the social context were significantly lower in the iso total session (t(39) = 2.68, $P = 0.009$) compared to the iso with media session. This indicates that, following total isolation, participants showed increased seeking of high rewards in the social context compared to isolation with access to social media.

We performed a sensitivity analysis to assess whether session order effects might have driven the changes in RTs (see details in Supplementary Note 3) and did not find any effect of session order on RTs in the EBDM task.

### Exploratory analyses: Isolation-induced changes in state loneliness and changes in social reward seeking.

We found a correlation between the difference in state loneliness and the difference in RTs in the social context (r(38) = $-0.452$; $p = 0.003$, CI = $-0.66$,$-0.13$; Fig. 4) but no significant correlation in the non-social context (r(38) = $-0.030$; $p = 0.851$, CI = $-0.32$,0.32, (BF$_{10}$) = 0.20; Fig. 4; correlations were significantly different: z = $-1.96$, $p = 0.025$). Thus, participants who showed a larger difference in state loneliness between the two isolation sessions also showed more social reward seeking. We found no such effect for boredom or mood (see Supplementary Note 4 for details).

### Adolescents show increased RL following isolation

**Exploratory analyses: effects of session on performance measures.** Participants showed lower accuracy in phases 2 and 4, which had reversed reward contingencies compared to the acquisition phase 1 (Fig. 5A; phase 2: $b = -0.11$, t(936) = $-1.98$, CI = $-0.21$,$-0.001$, $P = 0.048$; phase 4: $b = -0.20$, t(936) = $-3.67$, CI = $-0.30$,$-0.09$, $P < 0.001$)). Accuracy was numerically, but not significantly, lower in phase 3, when reward contingencies were the same as during acquisition ($b = -0.10$, t(936) = $-1.90$, CI = $-0.20$,0.003, $P = 0.057$). We did not find differences in accuracy between sessions (baseline vs iso total: $b = -0.06$, t(936) = $-1.06$, CI = $-0.16$,0.05, $P = 0.289$; iso with media vs iso total: $b = -0.05$, t(936) = $-1.0$, CI = $-0.16$,0.05, $P = 0.320$; Fig. 5B).

Participants showed faster RTs during the iso total session compared with baseline ($b = 0.19$, t(936) = 2.48, CI = 0.04,0.35, $P = 0.013$; Fig. 5C).

Analogous to our analysis approach for the EBDM task, we performed a sensitivity analysis to assess whether session order effects might have driven the changes in RTs (see details in Supplementary Note 5) and did not find any effect of session order on RTs during RL.

We also found lower perseverative errors in the iso total session compared to baseline ($b = 0.10$, t(117) = 2.46, CI = 0.02,0.19, $P = 0.015$) and compared to the iso with media session ($b = 0.08$, t(117) = 2.52, CI = 0.02,0.15, $P = 0.013$) for phase 2 (first reversal phase), across feedback type (Fig. 5D). We found no difference between sessions in perseverative errors for phase 3 (baseline vs iso total: $b = -0.003$, t(117) = $-0.06$, CI = $-0.09$,0.08, $P = 0.955$; iso with media vs iso total: $b = -0.03$, t(117) = $-0.60$, CI = $-0.14$,0.07, $P = 0.547$) and phase 4 (baseline vs iso total: $b = -0.02$, t(117) = $-0.48$, CI = $-0.12$,0.07, $P = 0.630$; iso with media vs iso total: $b = -0.01$, t(117) = $-0.29$, CI = $-0.10$,0.07, $P = 0.772$).

**Computational modelling.** We found that the FU model showed the best-fitting results controlling for model complexity (see methods and Supplementary Table 2). We then assessed and validated the model fit of the FU model by running posterior predictive checks and parameter recovery (see methods for details). Parameter recovery was successful, and the model predictions captured the data well (Supplementary Fig. 2).

To test the effects of isolation on RL (H3) and whether social media use during isolation would remediate effects of isolation on RL (H4), we extracted the population-level mean posterior distributions of each parameter estimated by the model: learning rates (learning_pos and learning_neg) and exploration (beta). Figure 6 depicts the mean posterior distributions for each model parameter (Fig. 6A) and the difference distributions for each case and each parameter (Fig. 6B).

Learning_pos rates. There was a 90% probability for a higher learning rate from social feedback (BF$_{10}$ = 3.29; 95% credible interval (CrI) of difference distribution = $-0.07$,0.26) and a 93% probability for higher learning from non-social feedback (BF$_{10}$ = 3.66; CrI = $-0.05$,0.29) in the iso total compared to the baseline session.

Learning_neg rates. There was a 96% probability for a higher learning rate from social feedback in the iso total compared to the baseline session (BF$_{10}$ = 5.30; CrI = $-0.01$,0.21) and a 99% probability for a higher learning rate from social feedback in the iso with media compared to the baseline session (BF$_{10}$ = 20.40; CrI=0.08,0.24).

Of note, confirming previous findings[72,73], learning rates from positive prediction errors were overall higher compared to learning rates from negative prediction errors.

Beta. There was an 88% probability for a higher beta (indicating less exploration) for social feedback (BF$_{10}$ = 7.29; CrI = $-0.13$,0.48) and an 86% probability for a higher beta for non-social feedback (BF$_{10}$ = 4.98; CrI = $-0.11$,0.38) in the iso total compared to the baseline session. Beta was also higher in the iso with media compared to the baseline session (social:100% probability, BF$_{10}$ = 45.33; CrI=0.24,0.73; non-social: 96% probability,

**Table 1 | Results pairwise comparisons T0 vs T3 during isolation for self-report measures of affect for both isolation sessions (iso total and iso with media; N = 40)**

| Measure (T0 vs T3) | t (df) | | p | | Effect size (Cohen's d) | |
|---|---|---|---|---|---|---|
| | Iso total | Iso w media | Iso total | Iso w media | Iso total | Iso w media |
| **State Loneliness** | 6.79 (39) | 4.50 (39) | <0.001 | <0.001 | 1.04 | 0.63 |
| **Positive Mood** | −6.23 (39) | −6.23 (39) | <0.001 | <0.001 | 0.74 | 0.74 |
| **Negative Mood** | −0.32 (38) | −0.68 (38) | 0.753 | 0.503 | 0.06 | 0.13 |
| **Boredom** | 8.32 (38) | 4.17 (39) | <0.001 | <0.001 | 1.45 | 0.65 |
| **Social Craving** | −0.94 (39) | −0.38 (38) | 0.353 | 0.709 | 0.13 | 0.06 |

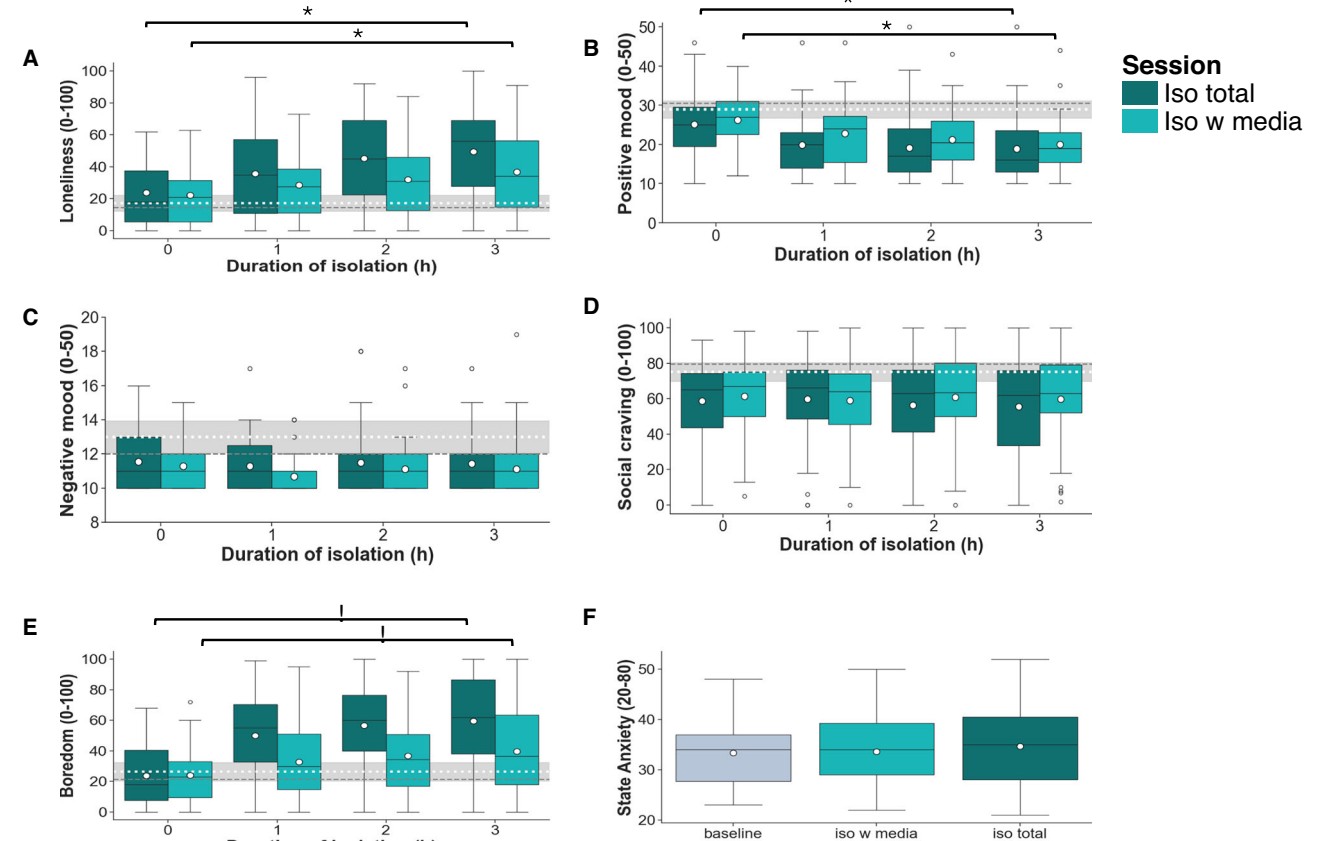

**Fig. 2 | Effects of isolation on self-report measures of affect.** Changes in self-reported state loneliness (**A**), positive mood (**B**), negative mood (**C**), social craving (**D**), boredom (**E**) over time during isolation. State anxiety (**F**) measures for each session (N = 40). Dark green = iso total session; turquoise = iso with media session. The boxplots indicate the median (black centre line), mean (white circle), the interquartile range (IQR; box) and the 1.5 x IQR minima and maxima (whiskers). The black rhombi indicate outliers (values outside of 1.5 x IQR). The dashed lines across the plots indicate the mean (white) and median (grey) ratings during baseline and their 95% confidence interval (grey area). * indicates p < 0.001.

BF$_{10}$ = 16.0; CrI = −0.06,0.65). In addition, beta was lower in the iso total compared to the iso with media session (social: 97% probability, BF$_{10}$ = 20.92; CrI = −0.65,0.03; non-social: 78% probability, BF$_{10}$ = 4.98; CrI = −0.53,0.25).

To interpret the parameter results more meaningfully, we used simulations to assess the optimal learning rate and inverse temperature for our specific task environment (see methods for details). As expected[65] for a short task (28 trials per block) with frequent changes of reward contingencies (every 7 trials), a high learning rate (indicating stronger reliance on more recent outcomes for value updating) was optimal (Supplementary Fig. 1). Thus, for the current task, frequently updating reward contingencies was associated with better performance.

For beta (inverse temperature), values > 3 were shown to be optimal. Note that participants in our task showed overall lower beta values,

indicating relatively high exploration (Fig. 6A), with most values ranging between 0.5–2 across sessions.

**Exploratory analyses: Isolation-induced changes in state loneliness and changes in social RL.** We found a correlation between the difference in state loneliness and the difference in learning_neg rates from social feedback (r(38) = 0.384; p = 0.015, CI = 0.05,0.62; Fig. 7) but no significant correlation for learning from non-social feedback (r(38) = 0.222; p = 0.169, CI = −0.11,0.50, BF$_{10}$ = 0.49; Fig. 7; correlations were not significantly different: z = 0.754, p = 0.225). Thus, participants who showed a larger difference in state loneliness between the two isolation sessions also showed higher RL from negative social feedback. We found no such effect for boredom or mood (see Supplementary Note 6 for details).

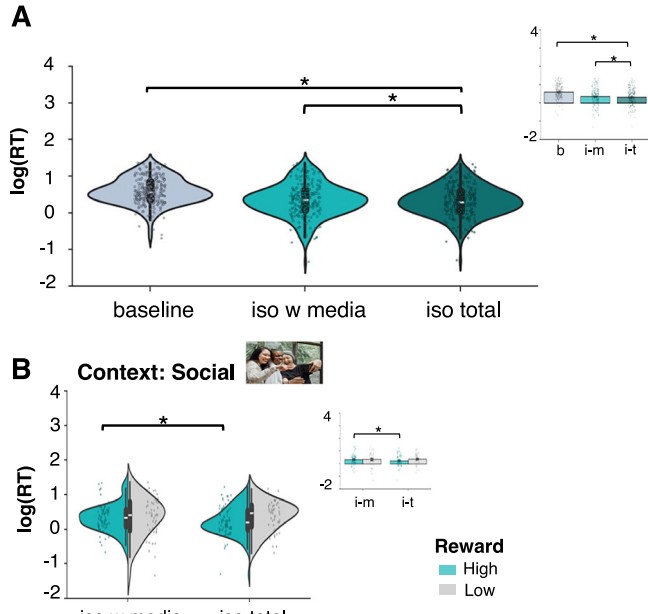

**Fig. 3 | Effects of isolation on reward seeking. A** Main effect session: Faster response times (RTs) during the reward seeking (EBDM) task following total isolation compared to baseline and compared to the iso with media session ($N = 40$). The violin plots depict log-transformed response times (RTs) plotted for each session and illustrate the distribution of the data. The insert bar plots depict the contrast values for each session: baseline (b), iso with media (i–m) and iso total (i–t), showing the mean for each session. **B** Interaction session x reward x context: Faster response times (RTs) during the reward seeking (EBDM) task following total isolation compared to the iso with media session for high reward trials in the social context. Violin plots depict log-transformed response times (RTs) plotted separately for high rewards (green) and low rewards (grey) for the social context. In each plot, the white horizontal lines indicate the median, the bold black vertical lines the IQR and the thin black vertical lines the $1.5 \times$ IQR minima and maxima. The error bars indicate the standard error of the mean. * indicates $p < 0.05$. Stimulus material displayed in 3B was taken from a freely available picture database (https://www.pexels.com/).

We conducted additional exploratory analyses on these data, which we report in the document *Additional exploratory analyses* on https://osf.io/qrhd6/ for full transparency.

## Discussion

In adolescents aged 16–19 years, short-term social isolation was associated with an increase in responsiveness to rewards measured in two different tasks. First, we found that, even after a relatively short period of isolation (between 3.5 and 4 h), participants reported feeling lonelier at the end of the isolation period than they did at the beginning. This was true following both total isolation and, to a lesser extent, isolation in which participants were able to interact with others virtually. Second, the results from the EBDM task showed that total isolation was associated with faster RTs compared to baseline and compared to the iso with media session when participants were deciding whether or not to undergo an effortful task. This was especially true for high rewards in the social context, which is indicative of increased social reward seeking[74]. Third, the results from the RL task revealed that participants showed better performance in the total isolation session (compared to baseline and compared to the iso with media session), indicated by lower perseverative errors during RL in that session. Furthermore, our modelling analysis showed that total isolation (but not isolation with social media) was associated with an increase in RL from positive feedback. Isolation was also associated with an increase in RL from negative social feedback and lower exploration (for both isolation sessions). Our findings are partially consistent with results from animal models, which have shown that social isolation in adolescent rodents increases reward responsiveness[26]. In

animals, social isolation increases responsiveness to social rewards[75], seeking of food or drug rewards and risk of developing addictions[28]. Animal models have also shown that isolation improves learning of stimulus-reward associations[30,31] (but not their reversal[32–34]). Our findings suggest that there is a similar effect of increased responsiveness to rewards immediately following total isolation in human adolescents. These findings are also in line with human research on chronic loneliness (in adults), showing that individuals high in loneliness show stronger responsiveness to social rewards[76–78].

While we found stronger reward responsiveness to social rewards following total isolation in both reward tasks, we also found evidence for increased reward seeking for non-social rewards and increased RL from positive feedback, following total isolation. This is in line with findings from animal research showing higher reward seeking following isolation in the non-social domain as well[28,75].

We found that individuals who reported higher state loneliness during isolation also showed stronger reward seeking. Specifically, the difference in subjective feelings of state loneliness between the two isolation sessions (which were two highly controlled active conditions) predicted the difference in reward seeking in the social context of the EBDM task, but there was no significant correlation in the non-social context. Thus, individuals who reported higher state loneliness also showed stronger social reward seeking following total isolation. We found a similar relationship for the difference in learning rates from negative feedback in the RL task, but the difference in correlations between social and non-social feedback was not significant for this task. While these results suggest that, in line with reward seeking, individuals who reported higher state loneliness following isolation also showed stronger RL after total isolation, we cannot conclude that this finding is specific for RL from social feedback (as opposed to RL from non-social feedback) as the difference in correlations between social and non-social feedback was not significant for this task.

We did not observe any such effects for changes in mood or boredom. The current results therefore suggest that there is a strong relationship between state loneliness and reward responsiveness. However, we cannot rule out that the observed effects might have been partially driven by other (unmeasured) variables, such as general arousal. Indeed, past research has shown that isolation leads to decreases in general arousal[18]. However, because low arousal has been associated with reduced motivation in past research[79], it is unlikely that changes in general arousal were driving our results. Nevertheless, future studies should include measures of arousal and reward responsiveness following isolation to explore any potential interaction between these effects.

The finding that total isolation was associated with improved reversal learning is contrary to results from animal studies[32–34]. Animal studies have found that, while isolation can improve initial learning of stimulus-reward associations[30,31], it leads to perseveration to initially learned reward-outcome associations following reversals[32–34]. In contrast, in our study, adolescents constantly updated associations between cues and rewards throughout the task, particularly in response to social feedback, and showed fewer perseverative errors following total isolation. Thus, our results suggest that, following total isolation, adolescents placed more weight on each instance of feedback they received to guide their behaviour, especially when that feedback was social. In everyday life, such a behaviour might lead to an overemphasizing of irrelevant features[65]. Specifically, while a high learning rate was optimal in our task (see Supplementary Fig. 1 for simulations), high learning rates are not always optimal and the interpretation of a learning rate depends on the specific task environment[65]. A high learning rate indicates that participants only use the most recent outcomes for value updating, while a low learning rate indicates that both recent outcomes and previous outcomes contribute to the value update[65]. Thus, a low learning rate can be optimal in task environments in which reward contingencies are more stable. Considering this, our results suggest that, following total isolation, adolescents place more weight on each instance of social feedback they receive to guide their behaviour. Furthermore, our finding of increased learning from negative social feedback (in addition to increased learning

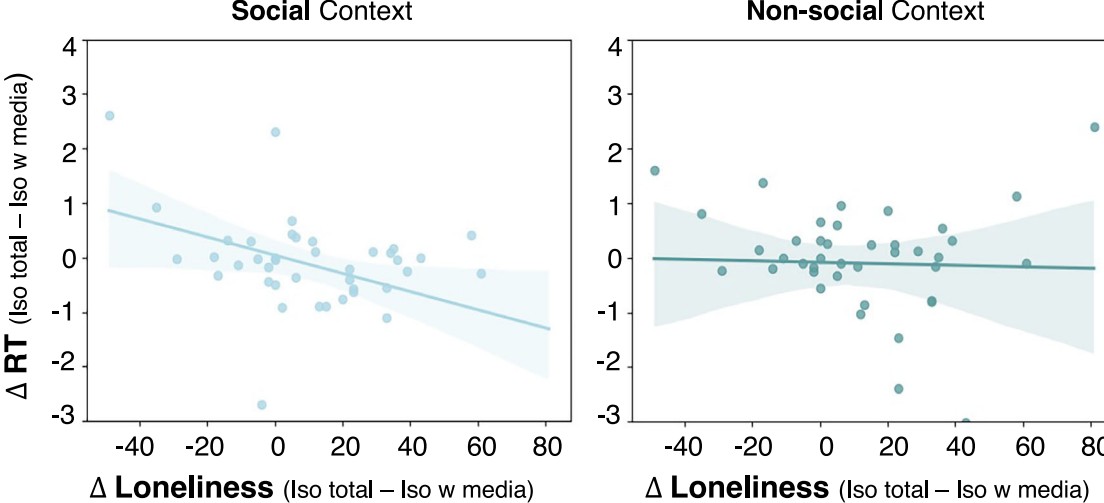

**Fig. 4 | Correlations between change in self-reported state loneliness and change in RTs in the EBDM task.** Scatter plots depict the difference between iso total and iso with media in self-reported state loneliness levels (after three hours of isolation) and RTs during the EBDM task for high reward, and high effort trials separately for the social and non-social context ($N = 40$). The scatter plots illustrate the negative correlation between the difference in state loneliness and the difference in RTs in the EBDM task in the social context and show individual data points and regression lines indicating the linear fit and the shaded areas depicting the 95% confidence interval. The plots show that a stronger increase in state loneliness in the iso total compared to iso with media session was associated with a larger decrease in RTs in the task for the social context.

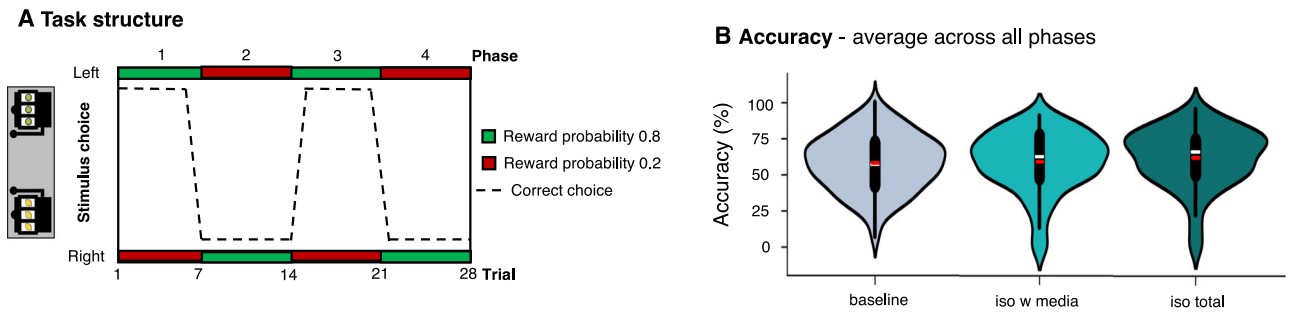

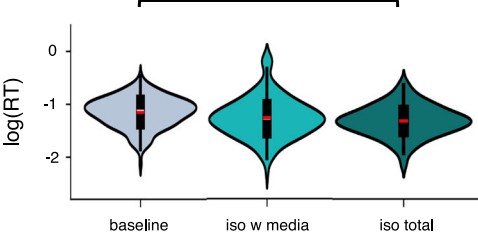

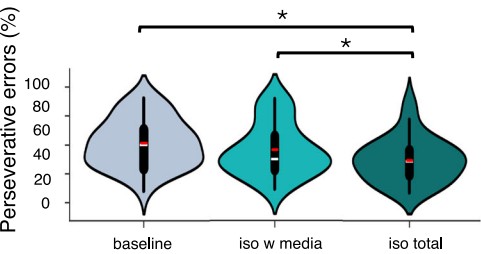

**Fig. 5 | Effects of isolation on reward learning – behavioural indices. A** The structure of the task. Feedback was given with an 80:20 reward ratio; green blocks indicate high reward (80%) and red blocks low reward (20%). Which stimulus was rewarded was reversed every 7 trials generating 4 phases of the task. **B** Average accuracy over the whole task across social and non-social blocks shown for each session (baseline, iso with media, iso total; $N = 40$). There was no difference in accuracy between sessions. **C** Average RTs over the whole task across social and non-social blocks for each session. Participants were faster in the iso total compared to the baseline session. **D** Perseverative errors for phase 2 (first reversal phase) across social and non-social blocks are shown for each session. Participants made fewer perseverative errors in the iso total compared to the iso with media session and to the baseline session. The violin plots illustrate the data distributions for the different conditions. The inserted boxplots illustrate the interquartile range (IQR; box) the white lines signify medians and the red lines means. The whiskers extend from the hinge to at most 1.5 times the interquartile range.

from positive social and non-social feedback) indicates that participants might also show hypervigilance to negative social cues following isolation, which is in line with findings from past research on chronic loneliness[80,81].

Participants reported feeling lonelier at the end of the total isolation period than they did at the beginning and, interestingly, this was also true when adolescents had access to virtual social interactions during isolation.

However, the increase in self-reported state loneliness was higher following total isolation. In contrast, positive mood decreased to a similar extent in both isolation sessions. Whether virtual social interactions fulfil social needs is debated[36] and some findings suggest the contrary, that is, that virtual social interactions can increase loneliness[36]. The current study showed that during isolation with access to virtual social interactions, participants self-reported

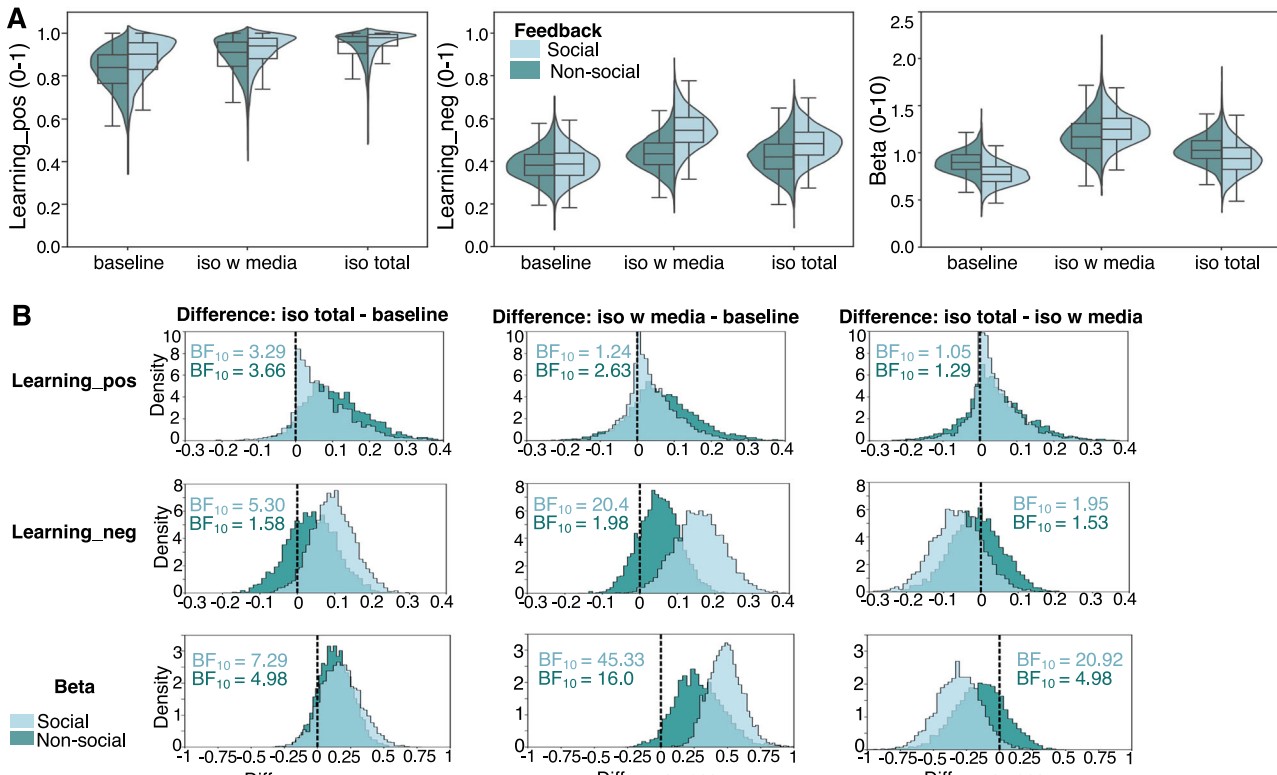

**Fig. 6 | Effects of isolation on reward learning – model results. A** Population-level mean posterior distributions of learning rates from positive prediction errors (learning_pos), learning rates from negative prediction errors (learning_neg) and inverse temperature (beta) for each session for each feedback type ($N = 40$). The violin plots illustrate the posterior distributions for the population-level mean effects of the different conditions. The posterior distributions consist of 12,000 samples combined from four Monte Carlo Markov chains. The boxplots illustrate the interquartile range (IQR; box) and the centre lines signify medians. The whiskers extend from the hinge to at most 1.5 times the interquartile range. **B** Difference in population-level mean posterior distributions for each pair of sessions (i.e., the difference in iso total – baseline, iso with media – baseline and iso total – iso with media). Plots show the distributions for each feedback type (social, non-social). Rows show the difference distributions for each parameter (learning_pos rates, learning_neg rates and beta). The histograms depict the density of difference scores; the red dashed line indicates zero (no difference). The Bayes Factor in support of the alternative hypothesis ($BF_{10}$) is shown for each pairwise comparison. $BF_{10} > 3$ can be considered as meaningful evidence in support of the alternative hypothesis (i.e., that posterior distributions are different from zero meaning there is a difference between the sessions in this comparison).

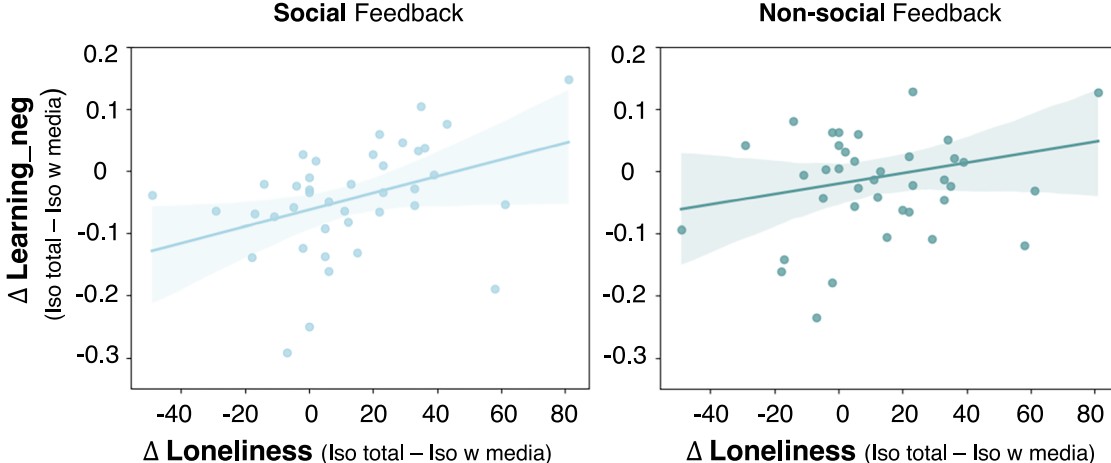

**Fig. 7 | Correlations between change in self-reported state loneliness and change in reward learning from negative feedback.** Scatter plots depict the difference between iso total and iso with media in self-reported state loneliness levels (after three hours of isolation) and Learning_neg rates from the RL task separately for Learning_neg rates from social and non-social feedback ($N = 40$). The scatter plots illustrate the negative correlation between the difference in state loneliness and the difference in learning_neg rates in the social context and show individual data points and regression lines indicating the linear fit and the shaded areas depicting the 95% confidence interval. The plots show that a stronger increase in state loneliness in the iso total compared to iso with media session was associated with a larger increase in learning_neg rates from social feedback.

lower feelings of state loneliness compared to total isolation. Thus, our findings provide support for the potential of social media to fulfil social needs. However, we note that in the present study, social isolation occurred only for a relatively short period of time, which could influence the effects of social media on fulfilling social needs. It is possible that the effects of social media may differ in the context of longer or prolonged periods of social isolation.

In line with this, in the iso with media session, adolescents showed lower reward seeking (in the EBDM task) compared to the isolation session and no significant difference to the baseline session. They also showed no difference in RL from positive prediction errors (in the RL task), compared with baseline (Fig. 6B, first row, column: Difference: is w media - baseline) and more perseveration compared to the isolation session (while there was no difference in perseveration compared to baseline; Fig. 5D). However, learning rates from negative prediction errors for social feedback were higher in both isolation sessions compared to baseline (Fig. 6B, second row, columns: Difference: is w media – baseline / Difference: iso total - baseline). This suggests that, following isolation (with and without access to social media), participants placed more weight on each instance of negative social feedback they received to guide their behaviour. This effect was stronger in participants who reported higher loneliness after isolation. We also found that inverse temperature was higher (indicating less exploration) in the iso with media session, especially on social feedback trials. Although lower exploration is thought to be adaptive in reinforcement learning tasks, higher exploration can also be adaptive in very dynamic environments[82]. Indeed, participants showed higher perseverative errors in the iso with media session compared to iso total (but not compared to baseline).

These findings could be driven by general effects of virtual interactions on attention or motivation. However, exploration (in line with learning_neg rates) was more strongly affected on social feedback trials suggesting a possible alternative explanation. A substantial number of participants (18 out of 40; 45%) reported spending more than 50% of their isolation time participating in virtual social interactions. Behaviour on social media platforms was shown to conform to the principles of RL: users space their posts to maximize the average rate of accrued social rewards, considering both the effort cost of posting and the opportunity cost of inaction[83]. Thus, a period of intense engagement in virtual social interactions might alter the reinforcing nature of other social cues, resulting in altered exploration toward such cues. However, this interpretation is speculative, and the findings require replication and further specification.

The present study raises key questions for future research. First, the neural processes underlying isolation-induced changes during reward seeking and RL remain to be explored. While increased activity in reward circuitry has been consistently shown in animal model studies of adolescent isolation[25,26], direct experimental assessment of reward circuit activity following isolation in humans would provide further evidence of isolation-induced increases in reward responsiveness.

Second, whether our experimental findings can be translated to the effects of real-life chronic (trait) loneliness is unknown. Chronic and state loneliness can have opposite effects on social behaviour[84]. The social homeostasis model suggests that prolonged engagement of mechanisms intended for short-term adaptation to state loneliness can cause pathological states during chronic loneliness[85]. This model was proposed based on research on the effects of isolation in animal models, but similar theoretical frameworks have been proposed based on human research on chronic loneliness[86]. For example, the cognitive control model of loneliness[86] similarly proposes that it is the prolonged engagement of regulatory mechanisms during loneliness that exhausts cognitive resources and results in aversive outcomes. Prolonged increases in reward responsiveness during chronic loneliness might ultimately result in imbalances of the reward system, increasing proneness to developing compulsive behaviours, or a blunting of reward system activity. Indeed, longitudinal studies in humans consistently link chronic loneliness to depression and substance use[7–16].

In the current study, we asked self-report questions about virtual social interactions during the iso-with-media session and report these in the results section. While our study was not sufficiently powered to assess how between-subject differences in social media use impacted the results for this session, this would be an important direction for future research. Furthermore, whether these findings are specific to adolescent participants or would also apply to other age groups should be tested in future studies.

## Limitations

A potential limitation of our study is that our experiment began with the baseline session for each participant. This design allowed us to examine the impacts of isolation and isolation with access to social media relative to an unaffected baseline. However, because the baseline session was first for each participant, order effects might have driven our results. We conducted sensitivity analyses to examine whether order effects might have driven our results (reported in the results section and described in detail in Supplementary Notes 3, 5) and did not find that order effects were driving the results. However, future research should counterbalance the sequence of all sessions.

The study was conducted between April 2021 and February 2022, which was during the Covid-19 pandemic. While participants were back to school full-time when this study was conducted, the unique context (i.e., general social distancing measuring still being in place and recent lockdowns) might have impacted adolescents' social behaviour and expectations, which could have affected the effects of social isolation.

Our comparison conditions to social isolation were a baseline session and a session where participants were isolated but had access to virtual social interactions. Future studies might explore alternative comparison conditions to the isolation session, such as in-person social interaction in the lab, for example, by providing confederates as interaction partners.

Whether the observed effects of isolation are specific to adolescents or would also be observed in other age groups remains to be explored. As our study does not allow any inferences about age specificity or developmental differences, future studies including different age groups would provide insight into age-related effects of social isolation.

## Conclusion

During adolescence, brain reward circuits undergo critical remodeling[87]. The domain of reward processing is of particular importance due to its powerful effects on behaviour[24] and mental health[88]. The present study suggests that brief durations of social isolation alter core components[24] of reward processing (i.e., reward seeking and RL) in human adolescents. Our findings indicate that real-life isolation (such as during a pandemic or when used as punishment in schools) might result in two key changes in adolescent behaviour. First, real-life isolation might result in higher seeking of rewards, such as food or recreational drugs. Second, the stronger reliance on each instance of feedback during RL following isolation suggests that real-life isolation might lead to an increased sensitivity to positive and negative feedback – especially social feedback - in adolescence.

## Data availability

Data are publicly available on the open science framework (OSF; https://doi.org/10.17605/OSF.IO/2M9XS (https://osf.io/2m9xs/). For the purpose of open access, the author has applied a Creative Commons Attribution (CC BY) license to any Author Accepted Manuscript version arising from this submission.

## Code availability

Analysis scripts and code used to generate the figures can be found on OSF https://doi.org/10.17605/OSF.IO/QRHD6 (https://osf.io/qrhd6/).

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

## Acknowledgements

This research was carried out at the Department of Psychology and the Wolfson Brain Imaging Centre at the University of Cambridge. The authors thank Michele Ferraro, Vicky Lupson and Tracy Horn for their support with data collection; Giacomo Bignardi and Richard Morey for discussions on methods; and Olivier Mougin, Catarina Rua and the UK7T Network (http://www.uk7t.org) for developing the original PSIR conversion script. We

gratefully acknowledge support of this project by a Henslow Research Fellowship from the Cambridge Philosophical Society (to L.T.). L.T. is also funded by Wellcome (Career Development Award). S-J.B. is funded by Wellcome (grant numbers WT107496/Z/15/Z and WT227882/Z/23/Z), the Jacobs Foundation, the Wellspring Foundation and the University of Cambridge Biomedical Research Centre. E.T. is funded by a Gates Cambridge Scholarship and a St. John's College Benefactors' Scholarship. K.T. is funded by a UKRI Economic and Social Research Council and the Gonville Research Studentship. S.P. is supported by the European Research Council (ERC; RaReMem: 101043804), and the Agence National de la Recherche (CogFinAgent: ANR-21-CE23-0002-02; RELATIVE: ANR-21-CE37-0008-01; RANGE: ANR-21-CE28-0024-01). L.Z. is supported by the University of Birmingham and a Wellcome Trust Data Science Ideathon Grant (228268/Z/23/Z). The computations described in this paper were partially performed using the Birmingham Environment for Academic Research (BEAR). The funders had no role in study design, data collection and analysis, decision to publish or preparation of the manuscript.

## Author contributions

L.T. and S-J.B. designed the study with input from E.T. and K.T.; L.T., E.T. and K.T. collected the data; L.T. analysed the data with support from S-J.B., S.P., L.Z., and E.T. L.T. wrote the manuscript, and all authors provided feedback on the final version.

## Competing interests

The authors declare no competing interests.
