## [Peer Review file · Communications Psychology]

Acute isolation is associated with increased reward seeking and reward learning in human adolescents

Corresponding Author: Dr Livia Tomova

Version 0:

Decision Letter:

Dear Dr Tomova,

Thank you for your patience during the peer-review process. Your manuscript titled "Acute isolation is associated with increased reward responsiveness in human adolescents" has now been seen by 3 reviewers, and I include their excellent comments at the end of this message. They find your work of interest but raised some important points. We are interested in the possibility of publishing your study in *Communications Psychology*, but would like to consider your responses to these concerns and assess a revised manuscript before we make a final decision on publication.

We therefore invite you to revise and resubmit your manuscript, along with a point-by-point response to the reviewers. Please highlight all changes in the manuscript text file.

Editorially, we would like to see a better integration of the neural findings, such as an explanation of the rationale for including neural measures and the formulation of clear hypotheses pertaining to the neural mechanisms in the introduction. Additionally, the interest of using a 7T could be further elaborated. From the reviewers' comments and our own reading, it is also clear that the manuscript would greatly benefit from a restructuring of the methods and results around a clearer narrative centered on a main thread. Moreover, we consider that the following aspects will need to be carefully addressed: (1) The concerns of reviewer #3 regarding the possible confounds of the test-retest effects; (2) The concerns of reviewer #1 regarding the representativity of the baseline in the context of restrictions linked to the COVID-19 pandemic, which raises some questions about how the baseline is different from the social isolation with media condition; (3) Reviewer #1 also raises some important concerns about the analysis of the RL task that we would like to see fully addressed; (4) Reviewer #2 raises the possibility of confounds such as the increased saliency of social cues after social isolation; (5) Finally, reviewers #2 and #3 raise important concerns on the interpretations of null effects and suggest alternative tests to draw conclusions without having to rely on them.

For manuscripts that interpret null results, we require Bayes Factors or equivalence tests to interpret the null results. Statements such as 'There is no difference between x and y.' or 'X does not affect Y.' must be revised to read 'We found [no/little] credible evidence of a difference between x and y.' or 'We found [no/little] credible evidence that X affects Y.'.

As reviewer #3 also suggested, it is our policy that authors must disclose all deviations from the preregistered protocol and explain the rationale for deviation (e.g., flaw, feasibility, suboptimality). In cases of deviation from the preregistered analysis plan for reasons other than fundamental flaw or feasibility, the originally planned analyses must also be reported. It is commendable that you already included such information in the supplementary information document. Please include relevant information in the main manuscript as well to facilitate the comparison between the preregistered protocol and the analysis. You can find our full policy on preregistration here: <https://www.nature.com/commspsychol/submit/preregistration>

I am attaching an Editorial Requests Table that details critical reporting requirements for the revised manuscript. Please attend to each item and ensure your manuscript is fully compliant. If your revised manuscript is not aligned with these requests on major issues, such as those concerning statistics, it may be returned to you for further revisions without re-

review.

Please submit the following items:

- Revised manuscript
- Point-by-point response to the referees' comments
- Cover letter (as a separate document)
- [Nature Research Reporting Summary](https://www.nature.com/documents/nr-reporting-summary.zip)
- [Editorial Policy Checklist](https://www.nature.com/documents/nr-editorial-policy-checklist.pdf)
- Completed Editorial Request Table (attached).

via this link: Link Redacted .

Additional guidance is available in our style and formatting guide [Communications Psychology formatting guide](https://www.nature.com/documents/commpsychol-style-formatting-guide-accept.pdf).

Best regards,

Eva R. Pool

Eva R. Pool, PhD
Editorial Board Member
Communications Psychology
orcid.org/0000-0001-5929-1007

REVIEWER EXPERTISE:

Reviewer #1: Social cognition, fMRI, decision-making

Reviewer #2: Social cognition, loneliness

Reviewer #3: Social cognition, fMRI, decision-making

REVIEWER REPORTS:

Reviewer #1 (Remarks to the Author):

The Authors present a well-designed experiment in which they explore how short-term isolation impacts self-reported loneliness, reward seeking behavior, and reward learning, using both behavior and fMRI in adolescents. The topic is certainly of interest to the field and I commend the authors for the sheer amount of work that they put into producing this manuscript. While the design of the experiment is largely sound, and the statistical analyses largely appropriate, some of the results are not entirely convincing and the organization/discussion of findings is difficult to follow. This interfered with assessing its potential larger contribution to the field. I would recommend major revisions prior to any acceptance. Below I detail specific questions and concerns that arose as I was reviewing the manuscript.

Overarching concern: Throughout the manuscript, there are a plethora of different analyses and results that are reported and

it is generally difficult to follow the main thread. That is to say that the organization and reasoning could be strengthened as it currently reads as a little unfocused and doesn't flow as easily as it could. That is not to say that the results are not interesting, and having too many findings is a great problem to have, but I found myself having to retrace and work hard to dissect the analyses of different subsets of data and the reasoning for some of the analysis choices. I realize that some of this is present in the supplement and/or the preregistration and I commend the authors on doing a thorough job, but I feel that the reader shouldn't have to track these things down as much as I found myself doing. This was also true in the Discussion, which could also benefit from some attention to flow (some paragraphs were only a sentence or two).

Specific Questions/Concerns:

1) Experimental Context: The pandemic period as a baseline for a loneliness paradigm is complicated. Multiple studies (e.g., Ernst et al, *Amer. Psych.*, 2022; Rusch et al., *Sci. Data*, 2023; Weissbourd et al., Making Caring Common Project report, 2021; etc.) suggest that loneliness was influenced by the COVID-Related restrictions in place in different countries, and further complicating things, many were relying on internet for social interaction when they hadn't necessarily in the past (thereby complicating the baseline for assessing the effects of having social media present). Indeed, the Authors do note in the methods that they had to adapt their inclusion/exclusion thresholds for social network sizes due to COVID-19. The exact dates during the pandemic the data were collected should be included, as well as what types of governmental restrictions were in place. The unique context of the time period in which the data were collected should also be considered in the discussion. This will provide reviewers/readers with a better sense of how to evaluate the data.

2) Intersession Durations: The authors should provide some more information on the duration between sessions (mean/median/SD).

3) Instructions in EDBM task: More clarity is needed concerning the instructions to participants. While it is clear in the figure of the experimental design that participants were instructed as to the condition type (Social/Non-social) in the EDBM task, it is not clear in the text.

4) Concerns about the Learning Task: It is not clear what participants knew about the structure of the learning task. If they were instructed that there were reversals, and what the probabilities of reward were, this would alter their behavior and change estimates of learning rate and softmax inverse temperature parameters. Examination of participant choice performance in figure 6A and supplementary figure 3 suggests that learning rates were (in my experience) quite high (and coming up against the upper boundary of 1 for learning_pos). This indicates that individuals were following outcomes quite closely, which raises the question of what they knew about the structure of the task. More information is needed to help the reviewer/reader evaluate the data.

5) Log Transforming of Reaction Times: In the EDBM task, analysis focuses on RTs, however, it doesn't seem from the description that RTs were log-transformed for statistical analyses. Log-transformation is recommended for RTs, not just because their distribution is typically positively skewed, but also because log-transforming of positive data is recommended to address assumptions of linearity and additivity (<https://statmodeling.stat.columbia.edu/2019/08/21/you-should-usually-log-transform-your-positive-data/>).

6) Fixed reversals in the learning task: I'm curious why reversals occurred every 7 trials without any jitter – across multiple reversals participants may begin to learn the structure of the task – which would impact estimates of learning rate and softmax inverse temperature parameters.

7) The learning task: From my reading of the description on pages 40-41, perseverative errors were defined as “two or more consecutive errors following reversal (following methods of Crawley et al. 2020)” and they were coded as binary per each reversal – i.e., 0 if there were no perseverative errors, 1 if there were two or more. I have some concerns. The first is that it seems that by simply coding the presence or absence of perseverative errors for each reversal, rather than coding for them on a trial-by-trial basis, the authors are losing information. I am unclear what the rationale for that would be. Second, the definition of two or more consecutive errors seems rather a low threshold unless participants know the probabilities and that there will be multiple reversals. Third – why not fit reversals separately from initial learning? I realize that the authors used an experience weighted model – but from my reading, that weighting simply decrements learning by a fixed rate over the full course of the experiment, which would be better suited for trait learning paradigm (in which target behavior is expected to be stable) or a single reversal paradigm. I'm not sure exactly how it decouples initial learning from reversal learning in this instance as stated by the authors. Finally, the weighted model doesn't contain separate learning rates for reward and punishment trials, making it a non-ideal comparison to the other two models.

8) Compensation for task behavior: It is unclear how participants were compensated for task behavior (i.e. whether it was based on a randomly selected trial or cumulative earnings). From the description of the EDBM task, it seems likely it was cumulative for that task, however, games were played for points and points were converted to monetary value at the end of the task. Were participants endowed with money in advance? Did they know what the conversion rate from points to currency was in advance? Description of how compensation was calculated should be included for each of the tasks that influenced compensation amounts.

9) Clarification of result reporting: In the results for the EDBM task, it seems like there are often different subsets of conditions being reported for different results (All RTs, RTs for High Effort trials, RTs for High-Effort/High-Reward trials) and it is not always clear which are being talked about or why the focus is on the particular data subset. The authors should review these results with an eye towards clarifying this and providing a clear rationale for each choice within the main text.

10) Absence of evidence is not evidence of absence: On page 17, line 356, the authors report “We found a correlation between difference in state loneliness and difference in learning_neg rates from social feedback ($r(38)=0.384$; $p=0.0145$; Figure 5) but not from non-social feedback ($r(38)=0.222$; $p=0.169$; Figure 5).” Looking at figure 7 just below this text, it seems likely that there is no statistical difference between these correlations. To avoid any confusion or over-interpretation, that should be directly tested and reported as the interpretation of this finding would differ where there a statistically significant difference between the social and nonsocial conditions.

11) fMRI univariate design matrix: According to the methods, “The design matrix included boxcars for the experimental conditions convolved with a double-gamma hemodynamic response function(HRF)”. Could the authors please clarify the duration of these boxcars?

12) fMRI univariate results and ROI analysis: The fMRI results add little to the manuscript in its current form. That said, I wonder more could be reported to further warrant their inclusion. First off, it would be appropriate to provide coordinates (in a table in the supplement) for the reward-sensitive regions identified in the contrast of High Reward > No Reward – as this would help for future meta-analyses and provide a sense of where exactly these regions were (e.g., the occipital cortex is a large region with many brain areas in it – where exactly were the reward-sensitive regions?). Second, it is unclear to me why the Iso+Media condition was completely excluded from the fMRI results. At the very least I would want to see whether there was a relationship between reward sensitivity in the OC and reward-seeking in the Iso+Media condition (i.e., as in figure 8B for the Iso Total and Baseline conditions). The lack of mention of the Iso+Media condition and lack of justification for its exclusion is perplexing. Finally, I wonder why the authors chose to use such a large ROI over the occipital cortex (at least from what I can tell from looking at the inset in Figure 8B)? What parts of the occipital cortex are covered with this ROI? It seems that it may cover retinotopic regions as well as lateral occipital regions. Why not focus the ROI analysis on those regions that were identified as reward sensitive within the ROI (i.e., a combined functional/anatomical ROI). This would ensure targeted and more interpretable results as well as likely boost the statistical power of the results.

13) Discussion of the fMRI results: on page 24 lines 513-514, the authors state: “individual differences in such attention signals to rewards might represent the neural mechanism underlying the contribution of the occipital cortex to the behavioural change.” This assumes that the occipital cortex IS contributing to behavioral change. Correlation with behavioral change is not the same as causally contributing to behavioral change, I would advise rewording.

Reviewer #2 (Remarks to the Author):

The paper presents a compelling investigation into the effects of isolation on reward sensitivity and social feedback processing in adolescents. Overall, the study is well-executed, with clear communication of its rationale, transparent methodology, and appropriate interpretations of the results. Below, I summarize the strengths of the paper and offer suggestions for some minor additions to the discussion section.

Strengths of the Paper

1. Clarity of rationale: The authors provide a clear rationale for their research questions and hypotheses. They draw on existing literature on adolescent social needs, isolation, and reward sensitivity to frame their study, making a persuasive case for examining the effects of short-term isolation on reward responsiveness. I think the theoretical framework is coherent and logically supports their hypotheses on the effect of isolation.
2. Transparency in methodology: The study’s methodology is described in sufficient detail to ensure reproducibility and facilitate replication. The authors clearly outline participant recruitment criteria, experimental procedures, task designs, and analytic strategies. I don’t know computational modelling very well so I couldn’t comment on the specific analyses but I appreciate the details in explaining the models. I also appreciate that the authors also clearly identified where they deviated from the original plan, for example, why they decided to use RT as outcomes due to low variance in choice data.
3. Clear communication of results: The results are presented clearly, with a distinction between findings that directly address the original research questions and those that are exploratory (e.g., correlation between difference in loneliness and difference in RT). The discussion section is thorough in relating results back to the original hypotheses and theoretical framework.
4. Appropriateness of interpretations: The interpretations of results are thoughtful and appropriately cautious, with the authors avoiding unwarranted causal claims and also being very clear about where the results do or do not generalize. They acknowledge that their study is primarily correlational and interpret their findings in light of this limitation.
5. Reflection on limitations: The authors thoughtfully discuss several limitations, including sample size, potential individual differences in baseline social connectivity, and the specificity of the fMRI findings to the occipital cortex. They also suggest meaningful directions for future research, which adds value to the paper.

Suggestions for Improvement

1. About the justification for why comparing total isolation with isolation with social media, the authors said: “This design allowed us to compare the effects between two tightly controlled and counterbalanced experimental sessions (iso total; iso with media), which differed only in the amount of social interactions participants had, while keeping other factors (such as spending time alone in a room) constant (see methods section for details on procedures)”. I believe the authors could further strengthen this justification by highlighting that this approach is also the most practical. An alternative design involving in-person interactions would require more logistical resources, including additional manpower (e.g., for an RA to act as interaction partner) and careful consideration of the nature of the interaction (e.g., structured interactions like the fast friend

paradigm, which has been used in prior studies as a control condition for solitude). Although, this can be a consideration for future research to consider alternative comparison to total isolation. In contrast, allowing participants to engage in social media provides a logistically feasible and scalable way to operationalize social interaction in a controlled manner, and keep it relatively constant across participant. Am I understanding the choice of iso w media correctly?

Do we see variance in the levels of social media engagement in the iso w media condition? The authors mentioned: "A substantial number of participants (18 out of 40; 45%) reported spending more than 50% of their isolation time participating in virtual social interactions" but do some not interact at all or do it very little?

2. While the authors interpret increased reward seeking in the social context as evidence of heightened reward sensitivity following isolation, what come to my mind is whether it is possible that this behavior might also reflect hypervigilance to social cues. There is a link between state loneliness and hypervigilance to social cues. Can the authors elaborate on this distinction in the discussion? Such as to tease apart what evidence in their findings would help us tease apart reward sensitivity versus hypervigilance?

For example, since hypervigilance often entails a stronger response to negative feedback or ambiguous social stimuli, this distinction could be further explored by comparing participants' responses to positive versus negative feedback; although I couldn't tell whether that is possible with this data. Such an analysis would help determine whether the observed behavior is better explained by heightened reward sensitivity (i.e., stronger learning from both positive and negative feedback) or hypervigilance (i.e., a stronger bias toward negative social cues).

3. The authors mention arousal briefly as a potential factor influencing reward responsiveness but do not elaborate on its role. Given the link between arousal and heightened sensitivity to environmental stimuli, including social cues, it would be valuable to discuss this more thoroughly in the discussion section.

Overall, I believe this paper makes a significant contribution to the understanding of how social isolation affects reward responsiveness in adolescents. The research is methodologically sound, the rationale is clear, and the results are appropriately interpreted.

Reviewer #3 (Remarks to the Author):

The present manuscript by Tomova and colleagues "Acute isolation is associated with increased reward responsiveness in human adolescents" investigated how short-term social isolation from personal interaction (with and without access to virtual social interaction) affected self-reported loneliness, reward seeking and reward learning adolescents of 40 young adolescents (16-19 years old). Participants reported to experience increased state loneliness with isolation (in general) as compared to baseline. Access to virtual interaction lead to reduced (state) loneliness reports. In the reward tasks, isolation was associated with faster decisions and "improved" reward learning. The study also employed fMRI and reported an association between lower neural reward sensitivity and "stronger effects of isolation". The authors conclude mainly that "isolation is associated with higher reward responsiveness".

Overall this manuscript is addressing a timely and relevant question in psychology regarding effects of experienced loneliness on important aspects of human cognition (here reward/motivation).

Strengths include the use of established behavioral tasks, an experimentally effortful isolation design (consistent with the first author's prior work), the adolescent age range and the preregistration of analyses.

Weaknesses exist with respect to manuscript structure, (potentially) interpretation of behavioral effects, lack of outlining specific hypotheses in the manuscript, social isolation condition framing, the motivation/justification for integrating neural data and some methodological/statistical aspects that deem further clarification.

Major comments:

Are the effects on reward task metrics clearly due to the isolation or study design (performing the same task twice). Could the RT effects in reward seeking after isolation for high reward conditions, as well as the RT effects in reward learning be explained by repetition effect (i.e. doing the task twice at baseline and again after isolation)- possibly not due to isolation? The authors may be able to address this (e.g. with prior literature) but it should be clearly stated in the manuscript especially if there is no specific effect between the isolation condition.

The introduction is very short, general (see next two comments) and lacks specific hypotheses. The manuscript would benefit from more details early on without forcing the reader to go specifically to the preregistration. Upon consultation of the SI and the preregistration document it becomes clear that more additional hypotheses were preregistered. That is not mentioned in the main manuscript and confusing. Please clarify that in the main manuscript briefly. Also, please clarify how/when it was decided which hypotheses were included in the manuscript.

Is social isolation with access to virtual interaction really "social isolation"? Especially in an age group where virtual social interaction may be an integral part of social interactions? Can the authors specify the respective operationalization of "social isolation" and maybe expand how this may relate to their prior work in adults and speculate on how difference in isolation types may vary in older adults?

Motivation for integrating neural data: the introduction is omitting sufficient background, rationalization and hypotheses for investigating brain function within the context of their study motivation. Overall the introductions seems very short and is lacking specific hypotheses.

Structure of the article: Consider reversing methods and results more in line with classic article structure, common also in this journal, to help the reader follow the necessary details and adding some less relevant details for understanding results to a SI (instead of a lengthy methods section at the end of the article)? If restructured the results section could be less "methods heavy" and help flow of the overall article. The methodological details in the results may be distracting from the actual results in the current article structure.

Did the Correlations between change in self-reported state loneliness and change in RTs in the EBDM task differ statistically between social and non-social feedback (figure 4)? The absence of an effect for non-social context does not imply a differing relationship otherwise (see <https://www.nature.com/articles/nn.2886>). I have the same question/concern for the relationship between difference in state loneliness and difference in learning_neg rates from social feedback btw social and non-social feedback, as well as the relationship between baseline iso total for PSC and RT

The title is implying a focus of the study on the neural data, but that does not reflect the emphasis thereof in the manuscript. Please clarify and adjust if necessary.

Can the authors comment in the discussion more on the fact that the only effect on neural reward sensitivity was in the occipital cortex – and not any other – potentially more expected region?

Minor comments:

Can the authors briefly specify more details on the non-standard processing of the 7T data or refer to the details in the preregistration? I could not find preprocessing details in the OSF code repository. This is relevant for reproducibility. The use of 7T in adolescents is relatively novel, if not at least (more) noteworthy. May there be any specific effects/confounds/opportunity for this study in 7T versus the more standard 3T?

Consider adding a definition of state loneliness earlier in the introduction.

Prior research on relationship between isolation and reward seeking is only briefly mentioned (and mainly the authors own work) in the introduction and the discussion.

10 social or virtual interactions seems low (at least without further reference). The authors say "frequent" but is there normative data that can be related to possibly? Similar comment for the chronic loneliness levels.

In the beginning of the discussion (pages 20/21/22) behavioral findings on reward behavior is stated as being consistent with findings in animals – but the link to human work should/could be highlighted more.

Methodological details that could/should be elaborated on:

- Specify fMRIPrep version.
- On pages 49/50, information on framewise motion exclusion criteria seems to contradict? (ART vs FMRIprep, FD vs FD&spike intensity?) Please clarify.
- What were the MRIQ reports used for? Was any data excluded based on IQMs?
- Please specify/approximate effect sizes for interactions.
- In the SI, please provide full statistical details (not just "all p values > 0.x")
- In SI – MRI data analysis: Instead of quoting sections from the preregistration that were executed as preregistered, consider describing what was done instead? Otherwise it is hard to decipher and potentially redundant to the methods section.

Version 1:

Decision Letter:

Dear Dr Tomova,

Thank you for your patience during the peer-review process, which took significantly longer than anticipated. One of the original reviewers withdrew, and although we were able to find a replacement who assessed the revisions, this added unexpected delays. Your manuscript titled "Acute isolation is associated with increased reward seeking and reward learning in human adolescents" has now been seen by our reviewers, whose comments appear below. In light of their advice I am delighted to say that we are happy, in principle, to publish a suitably revised version in *Communications Psychology*.

We therefore invite you to revise your paper one last time to address the remaining concerns of our reviewers and a list of editorial requests. At the same time we ask that you edit your manuscript to comply with our format requirements and to maximise the accessibility and therefore the impact of your work.

EDITORIAL REQUESTS:

Please follow Reviewer #4's suggestions to improve the readability and clarity of the results section. In addition, please include the follow-up analysis suggested by the reviewer, especially regarding the three-way interaction involving the effects of social isolation on reward seeking.

SUBMISSION INFORMATION:

OPEN ACCESS:

*** TRANSPARENT PEER REVIEW:** *Communications Psychology* uses a transparent peer review system. On author request, confidential information and data can be removed from the published reviewer reports and rebuttal letters prior to publication. If you are concerned about the release of confidential data, please let us know specifically what information you would like to have removed. Please note that we cannot incorporate redactions for any other reasons.

*** CODE AVAILABILITY:** All *Communications Psychology* manuscripts must include a section titled "Code Availability" at the end of the methods section. We require that the custom analysis code supporting your conclusions is made available in a publicly accessible repository at this stage; please choose a repository that generates a digital object identifier (DOI) for the code; the link to the repository and the DOI must be included in the Code Availability statement. Publication as Supplementary Information will not suffice.

* DATA AVAILABILITY:

All *Communications Psychology* manuscripts must include a section titled "Data Availability" at the end of the Methods section. More information on this policy, is available in the Editorial Requests Table and at <http://www.nature.com/authors/policies/data/data-availability-statements-data->

[citations.pdf](http://www.nature.com/authors/policies/data/data-availability-statements-data-citations.pdf)><http://www.nature.com/authors/policies/data/data-availability-statements-data-citations.pdf>.

Link Redacted

Best regards,

Troby Lui

Troby Lui, PhD
Associate Editor
Communications Psychology

REVIEWERS' COMMENTS:

Reviewer #1 (Remarks to the Author):

The authors did an excellent job of addressing the majority of my concerns and the paper reads very well – I have one minor comment but besides that I'm happy to recommend this for publication.

Minor Comment:

1) Page 13: L 403-405: "However, we were not able to run this analysis due to the low variance in participants' choice data. More specifically, many participants chose to play almost every trial in the task (73% of all trials were played)." 73% of trials seems to leave quite a bit of room for variance, perhaps the authors could include the between participant standard deviation for this metric? Alternatively, it would be helpful to know how many participants played 100% of trials.

Reviewer #4 (Remarks to the Author):

I was invited to comment on the revised manuscript without having reviewed the initial version of the paper. Specifically, I was asked to evaluate the changes made in response to Reviewer 3's comments.

Overall, I concur with the other reviewers that the manuscript addresses a timely, relevant, important, and scientifically valid research question on the impact of short-term social isolation on affect and reward processing in adolescents. I also share their enthusiasm regarding the quality of the research and its execution, the amount of work and effort that was dedicated to producing the manuscript, the clear description of the method, the calibrated interpretations, and the adoption of rigorous open-science practices promoting transparency. I particularly appreciated the inclusion of the supplementary note detailing the deviations from the preregistration in a clear and transparent manner, which is an excellent practice.

With respect to the revised manuscript, the authors have provided a clear and detailed revision that satisfactorily addressed most of the Reviewer 3's comments. The updated analytical approach (i.e., using the iso total session as reference rather than the baseline session) and the control analyses including session order converged in suggesting that the RT effects in the reward tasks are unlikely to be driven by practice/repetition effects, but more likely rely on the social isolation manipulation. The introduction now integrates well previous animal and human research on the effects of social isolation, clarifies how social isolation is defined, conceptualised, and operationalised, and clearly specifies the hypotheses.

These improvements notwithstanding, I believe there is still room to clarify and better streamline the presentation of the results. As it stands, the results section is somewhat difficult to follow. It is occasionally unclear which results correspond to which hypothesis, why some hypothesis-related results are reported in the supplementary information rather than in the main text, and why some results are presented in figures without being described in the text—despite being referenced in the discussion. These issues make it challenging to connect the discussion of the findings with the statistical results. It would also be important to clarify the extent to which the comparison between the two social isolation conditions (i.e., iso total vs. iso with media) is central to the hypotheses. Additionally, it would be beneficial to report in the main text how the sample size was determined, along with a justification. Some analytical decisions would also deserve to be more thoroughly justified. I elaborate on these points, along with other minor comments, in detail below. I hope these suggestions will be constructive and helpful in strengthening the manuscript even further.

Primary comments

1) The organisation of results section could be improved to make it easier for the reader to connect the statistical results with their interpretation in the discussion. As currently written, the results section is somewhat challenging to follow and integrate. It would be particularly helpful to clearly link the relevant results to the specific hypothesis being tested. This would better streamline the presentation of findings and clarify the extent to which they provide a compelling answer to the hypotheses.

2) In a similar vein, I think it would be important to report all primary analyses testing the hypotheses under consideration in the main text rather than in the supplementary information. This is particularly relevant for the hypothesis concerning the effects of social isolation on self-reported affect (i.e., increased loneliness, craving for social contact, state anxiety and negative mood, and decreased positive mood; H1), for which only the results on state loneliness and positive mood are currently included in the main text. My suggestion would be to (a) report the statistical results for all self-report dependent variables in a table, (b) briefly summarise the results in the main text, and (c) incorporate Supplementary Figure 3 into Figure 2. These changes would help present the results for H1 in a more comprehensive, concise, and accessible manner.

3) The results comparing the two social isolation conditions (i.e., iso total and iso with media) were not always consistently reported in the text, notably regarding the estimated parameters extracted from the computational models of the reward-learning task. It would be extremely helpful to clearly report these differences—even when they are not statistically significant—whenever they are relevant to the hypotheses, especially if they are discussed later in the manuscript. Doing so would help the reader better understand the connection between the modelling results (e.g., p. 32, ll. 918-931) and their interpretation.

4) Relatedly, it would be important to clarify the importance of the comparisons between the social isolation conditions in relation to the manuscript's overall contribution. Because these comparisons are not always consistently reported or discussed, it remains somewhat unclear whether they are central to the hypotheses. Expanding on this aspect would enhance the clarity of the manuscript and help the reader better situate and interpret its contributions.

5) Although the preregistration document clearly explains how the sample size was determined via an a priori power analysis based on pilot data, it would be highly beneficial to report this power analysis in the main text. Doing so would provide a transparent justification for the sample size (see Lakens, 2022) and make this key information readily available to the reader without requiring them to consult the preregistration. For completeness, it would also be warranted to specify the statistical test used in the power analysis to compute the required sample size (e.g., paired-sample t test).

6) Regarding the effects of social isolation on reward-seeking (see p. 23, ll. 685-689), I think it would be necessary to conduct a follow-up test to substantiate the interpretation that adolescents increased seeking of high rewards in the social context more after total isolation than after isolation with access to social media. While the three-way interaction between session, reward, and context suggests that the effect of reward level on reward-seeking depends on the social context—and that this interaction itself depends on the session—it does not clearly indicate which specific levels of the independent variables are driving the effect. A pairwise comparison directly testing RTs in the high-reward condition within the social context, contrasting total isolation with isolation with media, thus appears warranted to provide more direct evidence for the interpretation offered.

Secondary comments

7) On page 3 (ll. 98-99), the hypothesis (H4) concerning the modulatory effects of the social isolation with media includes only reward-seeking and reward learning. However, in the preregistration, it seems that this hypothesis (H7) also includes self-report measures of affect. If this is correct, it would be worth explaining or justifying this apparent discrepancy.

8) On page 4 in the participants section (l. 120), it would be appropriate to report the number of both adolescent women and men, as well as how sex/gender was determined, in order to align with the editorial policies of Communications Psychology.

9) Regarding the analysis of the self-report measures (see p. 12, ll. 356-360), it was unclear to me why only the measures taken at T0 and T3 (or T4, as reported in the supplementary information) were compared rather than including all timepoints in a single statistical model for each self-report measure. I think that a brief justification for these analytical specifications would be particularly helpful.

10) On page 14 (ll. 425-430) and page 21 (ll. 631-635), it may be worth specifying whether the variables used in the Pearson correlations were approximately normally distributed. If this was not the case, it would be advisable to additionally or alternatively report Spearman correlations.

11) In the results section and in the supplementary information, it would be beneficial to report the degrees of freedom associated with the t-statistics extracted from the linear mixed-effects models.

12) On page 25 (ll. 731-732), I would recommend avoiding characterizing the lower accuracy in phase 3 during the reward-learning task as a “trend”. Instead, I suggest interpreting all p-values above the alpha level as not statistically significant. The lower accuracy in phase 3 could be described as descriptively aligning with the pattern observed in phases 2 and 4, with the added caveat that this difference did not reach statistical significance.

13) On page 27 (ll. 774-774), it would be worthwhile to cite Palminteri and Lebreton (2022) with respect to the asymmetry between learning rates for positive versus negative prediction errors. This paper presents robust evidence for such asymmetry across different contexts and species.

14) On page 32 (ll. 915-916), the interpretation that the “findings provide support for the potential of social media to fulfil social needs” could benefit from further nuance. Specifically, it would be helpful to acknowledge the fact that social isolation occurred for only relatively short periods of time, which could influence the effects of social media on fulfilling social needs. It remains possible that the effects of social media may differ in the context of longer or prolonged periods of social isolation. Including this caveat would contribute to a more calibrated and context-sensitive interpretation of the potential effects of social media during social isolation.

Signed,
Yoann Stussi

References

- Lakens, D. (2022). Sample size justification. *Collabra: Psychology*, 8(1), Article 33267.

<https://doi.org/10.1525/collabra.33267>

- Palminteri, S., & Lebreton, M. (2022). The computational roots of positivity and confirmation in reinforcement learning. *Trends in Cognitive Sciences*, 26(7), 607-621. <https://doi.org/10.1016/j.tics.2022.04.005>

Response to reviewers

Table of Contents

Reviewer #1	2
Reviewer #2	11
Reviewer #3	14

Reviewer #1

Comment 1 - The Authors present a well-designed experiment in which they explore how short-term isolation impacts self-reported loneliness, reward seeking behavior, and reward learning, using both behavior and fMRI in adolescents. The topic is certainly of interest to the field and I commend the authors for the sheer amount of work that they put into producing this manuscript. While the design of the experiment is largely sound, and the statistical analyses largely appropriate, some of the results are not entirely convincing and the organization/discussion of findings is difficult to follow. This interfered with assessing its potential larger contribution to the field. I would recommend major revisions prior to any acceptance. Below I detail specific questions and concerns that arose as I was reviewing the manuscript.

Overarching concern: Throughout the manuscript, there are a plethora of different analyses and results that are reported and it is generally difficult to follow the main thread. That is to say that the organization and reasoning could be strengthened as it currently reads as a little unfocused and doesn't flow as easily as it could. That is not to say that the results are not interesting, and having too many findings is a great problem to have, but I found myself having to retrace and work hard to dissect the analyses of different subsets of data and the reasoning for some of the analysis choices. I realize that some of this is present in the supplement and/or the preregistration and I commend the authors on doing a thorough job, but I feel that the reader shouldn't have to track these things down as much as I found myself doing. This was also true in the Discussion, which could also benefit from some attention to flow (some paragraphs were only a sentence or two).

Reply 1: We thank the reviewer for their positive evaluation of our work and for the constructive feedback on how to improve the clarity of the manuscript. We have revised the organisation of the manuscript accordingly.

Specifically, we have now restructured and streamlined the manuscript to focus on a clear main thread. To do this, we have condensed analyses and removed some of the exploratory analyses that we do not consider central to the main research question (which we had originally reported in the supplementary materials and referred to in the main text in the results and discussion section). Some of the exploratory analyses effectively assessed the same question with slightly different methods and showed similar results (for example, if changes in self-reported state loneliness interact with the effects of isolation). While we initially considered the converging evidence from different analyses to be a strength of the manuscript, the reviewer's comment made us reconsider these analyses from the viewpoint of a reader and we agree that the multitude of analyses adds confusion rather than clarity. We have therefore decided to condense the exploratory analyses in the manuscript. For analyses that assessed the same question with slightly different statistical tests, we selected the analysis that we considered best suited and removed redundant analyses. Exploratory analyses that were not redundant, but also not central to the main research question (e.g. exploratory analyses on gender effects, or interactions between effects of isolation and chronic loneliness, etc.) were moved to a separate document, which we uploaded to OSF (*Additional exploratory analyses*). We did this to also streamline the Supplement and only focus on the essential analyses, while still being fully transparent in reporting all analyses we ran. We refer to the additional exploratory analyses in the results section of the main manuscript:

Revision – Manuscript page 30

“We conducted additional exploratory analyses on these data, which we report in the document *Additional exploratory analyses* on <https://osf.io/qrhd6/> for full transparency.”

Comment 2 - Experimental Context: The pandemic period as a baseline for a loneliness paradigm is complicated. Multiple studies (e.g., Ernst et al, Amer. Psych., 2022; Rusch et al., Sci. Data, 2023; Weissbourd et al., Making Caring Common Project report, 2021; etc.) suggest that loneliness was influenced by the COVID-Related restrictions in place in different countries, and further complicating things, many were relying on internet for social interaction when they hadn't necessarily in the past (thereby complicating the baseline for assessing the effects of having social media present). Indeed, the Authors do note in the methods that they had to adapt their inclusion/exclusion thresholds for

social network sizes due to COVID-19. The exact dates during the pandemic the data were collected should be included, as well as what types of governmental restrictions were in place. The unique context of the time period in which the data were collected should also be considered in the discussion. This will provide reviewers/readers with a better sense of how to evaluate the data.

Reply 2: We thank the reviewer for raising this important issue. Data collection started in April 2021 and finished on 16th February 2022. Participants were back in school full time when this study was being conducted. They might have been socialising slightly less than 'usual' but they were spending around 7 hours every weekday with peers. This information has been added in the methods section:

Revision – Manuscript page 5

“Data collection started in April 2021 and finished in February 2022. There were social distancing rules in place but no lockdowns or school closures during data collection. Thus, during data collection, young people were back in school full time and had regular social interactions.”

We also included a section about the unique context in the limitation section of the discussion:

Revision – Manuscript pages 34-35

“The study was conducted between April 2021 and February 2022, which was during the Covid-19 pandemic. While participants were back to school full time when this study was conducted, the unique context (i.e., general social distancing measures still being in place and recent lockdowns) might have impacted adolescents’ social behaviour and expectations, which could have affected the effects of social isolation.”

Comment 3 - Intersession Durations: The authors should provide some more information on the duration between sessions (mean/median/SD).

Reply 3: We have added this information to the section “Experimental procedures”:

Revision – Manuscript page 5

“...days between sessions ranged from 2 to 125 (M = 32.5, s.d. = 27.51) across participants”

Comment 4 - Instructions in EDBM task: More clarity is needed concerning the instructions to participants. While it is clear in the figure of the experimental design that participants were instructed as to the condition type (Social/Non-social) in the EDBM task, it is not clear in the text.

Reply 4: Thank you for catching this omission. We have added this information in the methods section:

Revision – Manuscript page 10

“At the beginning of each trial, participants were informed whether that trial was a social or non-social trial (Figure 1).”

Comment 5 - Concerns about the Learning Task: It is not clear what participants knew about the structure of the learning task. If they were instructed that there were reversals, and what the probabilities of reward were, this would alter their behavior and change estimates of learning rate and softmax inverse temperature parameters. Examination of participant choice performance in figure 6A and supplementary figure 3 suggests that learning rates were (in my experience) quite high (and coming up against the upper boundary of 1 for learning_pos). This indicates that individuals were following outcomes quite closely, which raises the question of what they knew about the structure of the task. More information is needed to help the reviewer/reader evaluate the data.

Reply 5: Thank you for highlighting the lack of clarity regarding this. We did not inform participants about the structure of the task (i.e., that there would be reversals or the probabilities of reward). As shown in Figure 5B, participants performed with on average 60% accuracy (std=15%) in the task (no sign. difference between sessions), which indicates that performance was not exceptionally high. For example, in the study by Crawley et al. 2018, who used the same probabilities of reward, adolescents reached an average accuracy of 76% (std = 14%; but note that their task only included one reversal while we had three, making our task slightly harder (Izquierdo et al. 2017)). Furthermore, in our revised analysis (in which we changed the predictor phase to a factor instead of a continuous predictor, which allows us to test for any relationship between block and accuracy instead of just a linear one with phase number), we find lower accuracy with increasing phase number in each session. Thus, participants became less accurate throughout the task. This further indicates that participants did not learn about the structure of the task, as we would expect increasing accuracy if they had.

However, it is correct that the learning_pos rates were high and near the upper boundary of 1. A high learning rate indicates that participants update their choices frequently and based on the most recent outcomes. Whether high or low learning rates are more optimal for performance depends on the specifics of the learning environment (see Zhang et al. 2020 for a discussion of how to interpret learning rates in reinforcement learning models).

We have now added more information about the information participants received about the structure of the learning task in the methods section:

Revision – Manuscript page 11

“Participants were not informed about the structure of the learning task (i.e., that there would be reversals) or the probabilities of reward. Participants were only told that their task was to find out, by trial and error, which is the more advantageous slot machine and win as many points as possible across the whole task.”

We have also included a section on the interpretation of learning rates in the discussion:

Revision – Manuscript pages 32

“Specifically, while a high learning rate was optimal in our task (see Supplementary Figure 1 for simulations), high learning rates are not always optimal, and the interpretation of a learning rate depends on the specific task environment⁶⁵. A high learning rate indicates that participants only use the most recent outcomes for value updating, while a low learning rate indicates that both recent outcomes and previous outcomes contribute to the value update⁶⁵. Thus, a low learning rate can be optimal in task environments in which reward contingencies are more stable. Considering this, our results suggest that, following total isolation, adolescents place more weight on each instance of social feedback they receive to guide their behaviour.”

Comment 6 - Log Transforming of Reaction Times: In the EBDM task, analysis focuses on RTs, however, it doesn't seem from the description that RTs were log-transformed for statistical analyses. Log-transformation is recommended for RTs, not just because their distribution is typically positively skewed, but also because log-transforming of positive data is recommended to address assumptions of linearity and additivity (<https://statmodeling.stat.columbia.edu/2019/08/21/you-should-usually-log-transform-your-positive-data/>).

Reply 6: We thank the reviewer for pointing us towards this discussion. We have now log-transformed reaction time data before running the linear mixed effect models as recommended in the link provided by the reviewer. The results of our analyses in the EBDM task have somewhat changed but still support our hypothesis of increased reward seeking following isolation. Specifically, we find that participants show faster RTs in the iso total session compared to baseline and compared to the iso with media session. We also find an interaction between session * context * reward, showing that participants have faster RTs after total isolation compared to isolation with social media in the social context when rewards were high. We have revised the results sections (and Figure 3) to reflect this change and have

also removed Supplementary Figure 1, which illustrated the 4-way interaction found in our previous analysis.

Revision – Manuscript pages 23-24

There was a main effect of session showing lower RTs in the total iso session compared to baseline ($b=0.32$, $t=5.06$, $CI=0.19,0.44$, $P<0.001$) and compared to the iso with media session ($b=0.16$, $t=2.55$, $CI=0.04,0.28$, $P=0.011$; Figure 3A). There was also a session * reward * context interaction showing lower RTs in the iso total compared to iso with media session for high reward trials in the social context (Figure 3B). See Supplementary Table 1 for full results. This indicates that, following total isolation, participants showed increased seeking of high rewards in the social context compared to isolation with access to social media.

FIGURE 3 Effects of isolation on reward seeking.

A. Main effect session: Faster response times (RTs) during the reward seeking (EBDM) task following total isolation compared to baseline and compared to the iso with media session. The violin plots depict log-transformed response times (RTs) plotted for each session and illustrate the distribution of the data. The insert bar plots depict the contrast values for each session: baseline (b), iso with media (i-m) and iso total (i-t), showing the mean for each session.

B. Interaction session x reward x context: Faster response times (RTs) during the reward seeking (EBDM) task following total isolation compared to the iso with media session for high reward trials in the social context. Violin plots depict log-transformed response times (RTs) plotted separately for high rewards (green) and low rewards (grey) for the social context.

In each plot, the white horizontal lines indicate the median, the bold black vertical lines the IQR and the thin black vertical lines the 1.5x IQR minima and maxima. The error bars indicate the standard error of the mean. * indicates $p < 0.05$.

We also log-transformed RTs in the RL task before including them in the mixed effects models to follow the same analysis procedure for each task. Here, we find the same result as in the previous analysis (faster RTs in the iso total session compared to baseline during reward learning; reported on page 26).

For the exploratory analyses testing whether differences in loneliness are correlated with differences in RTs between the iso total and iso with media session, we used the original (non-log-transformed) RT data because here we did not use linear models to analyse the data (but simply calculated difference scores between sessions and tested if they were correlated). We chose to run this analysis on the original data because that substantially simplified the interpretation of the results (i.e., subtraction scores from all-positive data are straightforward to interpret). We explain this approach in the methods section:

Revision – Manuscript page 14

“For this analysis, we used the original (non-log-transformed) data because recommendations from Gelman & Hill 2006⁴³ are for linear models while here we simply calculated difference scores between sessions and tested if they were correlated.”

Comment 7 - Fixed reversals in the learning task: I’m curious why reversals occurred every 7 trials without any jitter – across multiple reversals participants may begin to learn the structure of the task – which would impact estimates of learning rate and softmax inverse temperature parameters.

Reply 7: Thank you for this comment. For serial reversal learning paradigms, different structures have been used, including rule reversal after a fixed determined number (as in our task), a proportion of correct responses, or occurring with a probability after the criterion is fulfilled (see e.g., Izquierdo 2017, Yaple 2019 for reviews). We kept reversals fixed following a previous version of this task that we adapted for this study (Palminteri et al. 2017), as preregistered (see Variables: Reward learning in <https://osf.io/az3re>).

As stated above, our results show that accuracy decreased throughout the learning task in each session, indicating that participants did not learn the structure of the task, as we would expect increasing accuracy if they had.

Comment 8 - The learning task: From my reading of the description on pages 40-41, perseverative errors were defined as “two or more consecutive errors following reversal (following methods of Crawley et al. 2020)” and they were coded as binary per each reversal – i.e., 0 if there were no perseverative errors, 1 if there were two or more. I have some concerns. The first is that it seems that by simply coding the presence or absence of perseverative errors for each reversal, rather than coding for them on a trial-by-trial basis, the authors are losing information. I am unclear what the rationale for that would be. Second, the definition of two or more consecutive errors seems rather a low threshold unless participants know the probabilities and that there will be multiple reversals.

Reply 8: We thank the reviewer for raising this issue. While we based our approach for measuring perseveration in the task on past research (Crawley et al. 2020), we agree with the reviewer that choosing to label 2 or more consecutive errors as perseveration is a low threshold. Following the reviewer’s recommendation, we have chosen to implement a trial-by-trial coding of perseverative errors, which does not rely on predefined assumptions about whether perseveration occurred but simply counts the number of errors after reversal (which has also been used previously, e.g., Jentsch 2002). Using this approach, we find the same effect as in our original analysis (i.e., fewer perseverative errors after total isolation compared to isolation with social media after the first reversal in the task). We now also find a significant difference between the iso total session and baseline in this analysis.

We have updated the methods section to describe our new approach:

Revision – Manuscript page 15

“Perseverative errors were quantified as the proportion of errors choosing the previously reinforced stimulus following each reversal.”

We updated the results:

Revision – Manuscript page 26

We also found lower perseverative errors in the iso total session compared to baseline ($b=0.10$, $t=2.22$, $CI=0.01,0.19$, $P=0.026$) and compared to the iso with media session ($b=0.08$, $t=2.06$, $CI=0.004,0.16$, $P=0.040$), across feedback type (Figure 5D).

And we updated Figure 5D:

FIGURE 5 Effects of isolation on reward learning – behavioural indices.

A. The structure of the task. Feedback was given with an 80:20 reward ratio; green blocks indicate high reward (80%) and red blocks low reward (20%). Which stimulus was rewarded was reversed every 7 trials generating 4 phases of the task. B. Average accuracy over the whole task across social and non-social blocks shown for each session (baseline, iso with media, iso total). There was no difference in accuracy between sessions. C. Average RTs over the whole task across social and non-social blocks for each session. Participants were faster in the iso total compared to the baseline session. D. Perseverative errors for phase 2 (first reversal phase) across social and non-social blocks are shown for each session. Participants made fewer perseverative errors in the iso total compared to the iso with media session and to the baseline session. The violin plots illustrate the data distributions for the different conditions. The inserted boxplots illustrate the interquartile range (IQR; box) the white lines signify medians and the red lines means. The whiskers extend from the hinge to at most 1.5 times the interquartile range.

Comment 9 - Why not fit reversals separately from initial learning? I realize that the authors used an experience weighted model – but from my reading, that weighting simply decrements learning by a fixed rate over the full course of the experiment, which would be better suited for trait learning paradigm (in which target behavior is expected to be stable) or a single reversal paradigm. I'm not sure exactly how it decouples initial learning from reversal learning in this instance as stated by the authors.

Reply 9: We thank the reviewer for this comment. We do not believe that participants were processing the pre and post reversal phase as two clearly distinct phases that warrant a separate set of model parameters. The purpose of computational modelling is to *approximate* cognitive processes. Given that our participants were never informed/cued about the change in reward probabilities, the models should not process this information either (Lewandowsky & Farrell 2010; Zhang et al. 2020).

Regarding the EWA model, although the learning rate (ϕ) is fixed overall, value update does depend on the experience updating (n), which is changing in a trial-by-trial manner. Regardless, our statement regarding the decoupling of initial learning from reversal learning was taken from past research using this model (de Ouden et al 2013) and this study was indeed a single reversal paradigm. We agree with the reviewer that, in the case of a serial reversal learning paradigm, the interpretation is more difficult and we have removed the statement regarding the decoupling of initial and reversal learning now. In any case, the model did not fit our data well and we did not use it for any further analyses.

References:

Lewandowsky, S., & Farrell, S. (2010). Computational modeling in cognition: Principles and practice. Sage.

Zhang, L., Lengersdorff, L., Mikus, N., Gläscher, J., & Lamm, C. (2020). Using reinforcement learning models in social neuroscience: frameworks, pitfalls and suggestions of best practices. *Social Cognitive and Affective Neuroscience*, 15(6), 695-707.

Comment 10 - Finally, the weighted model doesn't contain separate learning rates for reward and punishment trials, making it a non-ideal comparison to the other two models.

Reply 10: Thank you for catching this omission. We have now included an EWA model with separate learning rates for reward and punishment trials in the model comparison. We kept the original model (with one learning rate) in the model comparison as this was the model we had preregistered. The updated model with separate learning rates did not outperform our original winning model (fictitious update model with two learning rates). Meanwhile, our model-based results (given that there was no change of the winning model) remain the same.

We have revised the methods section to include the new model:

Revision – Manuscript page 17

(iii) The experience-weighted attraction model with separate learning rates for positive and negative prediction errors.

$$V_{c,t} = \begin{cases} (V_{c,t-1} \times \varphi^{\text{rew}} \times n_{c,t-1} + O_{t-1})/n_{c,t}, & \text{if } O_{t-1} > 0 \\ (V_{c,t-1} \times \varphi^{\text{nrew}} \times n_{c,t-1} + O_{t-1})/n_{c,t}, & \text{if } O_{t-1} < 0 \end{cases} \quad (4)$$

And we revised Supplementary Table 2 to include the new model (“EWArp”):

Revision – Supplementary Information page 12

Supplementary Table 2. Results of model comparison for RL task.

Model	LOOIC	Bayesian stacking	Pseudo-BMA	WAIC
RP	6496.026	0.137	0.033	6420.713
EWA	6547.515	0.000	0.000	6445.719
EWArp	6510.420	0.330	0.025	6418.882
FU	6388.746	0.533	0.942	6315.901

We report the following model comparison metrics: LOOIC = Leave-One-Out Information Criterion; Bayesian Stacking = stacking of means¹; Pseudo-BMA = Pseudo Bayesian Model Averaging¹; WAIC = Widely Applicable Information Criterion; RP = Reward-Punishment Model; EWA = Experience-Weighted Attraction Model; EWArp = Experience-Weighted Attraction Model with separate learning rates for positive and negative prediction errors; FU = Fictitious Update Model. For LOOIC and WAIC, lower values indicate better model fit. For Stacking and Pseudo-BMA higher values indicate better model fit. We used Bayesian Stacking and Pseudo-BMA as model weight metrics to determine the winning model (see methods for details). The Fictitious Update Model is marked grey as it shows the best model fit.

¹Yao, Y., Vehtari, A., Simpson, D. & Gelman, A. Using Stacking to Average Bayesian Predictive Distributions (with Discussion). *Bayesian Anal.* 13, 917–1007 (2018).

Comment 11 - Compensation for task behavior: It is unclear how participants were compensated for task behavior (i.e. whether it was based on a randomly selected trial or cumulative earnings). From the description of the EBDM task, it seems likely it was cumulative for that task, however, games were played for points and points were converted to monetary value at the end of the task. Were participants endowed with money in advance? Did they know what the conversion rate from points to currency was in advance? Description of how compensation was calculated should be included for each of the tasks that influenced compensation amounts.

Reply 11: We have included more information on this in the methods section:

Revision – Manuscript page 10

“Each point equalled 2 pence and participants received the cumulative earnings they made across the whole task (which was up to £2.50). Participants were told that each point they won on the task would translate to actual money that they would receive in addition to their participant reimbursement. Participants were not informed about the exact conversion rate but were told that the combined additional wins for both reward tasks in each session could be up to £5. There was no endowment with money in advance as the task did not involve any losses.”

Comment 12 - Clarification of result reporting: In the results for the EBDM task, it seems like there are often different subsets of conditions being reported for different results (All RTs, RTs for High Effort trials, RTs for High-Effort/High-Reward trials) and it is not always clear which are being talked about or why the focus is on the particular data subset. The authors should review these results with an eye towards clarifying this and providing a clear rationale for each choice within the main text.

Reply 12: We thank the reviewer for this important comment. Our main, preregistered, analysis tested the effects of isolation on RTs for the different combinations of effort and reward: high effort - high reward; low effort– high reward; high effort – low reward; and low effort - low reward. Our exploratory analyses focused only on high effort-high reward trials as we consider these the most relevant for studying motivation and reward seeking (i.e., capturing participants’ willingness to expend effort for rewards see Treadway et al. 2009 for a discussion on this topic). However, we agree with the reviewer that this approach was not explained well. We have added information about which analyses were preregistered, and which were exploratory and a rationale for choosing to run exploratory analyses only on high reward high effort trials in the methods section:

Revision – Manuscript page 14

“We focused on high reward- high effort trials in these analyses as we consider these the most relevant for studying motivation and reward seeking (i.e., capturing participants’ willingness to expend effort for rewards see ³⁶ for a similar rationale)”.

Comment 13 - Absence of evidence is not evidence of absence: On page 17, line 356, the authors report “We found a correlation between difference in state loneliness and difference in learning_neg rates from social feedback ($r(38)=0.384$; $p=0.0145$; Figure 5) but not from non-social feedback ($r(38)=0.222$; $p=0.169$; Figure 5).“ Looking at figure 7 just below this text, it seems likely that there is no statistical difference between these correlations. To avoid any confusion or over-interpretation, that should be directly tested and reported as the interpretation of this finding would differ where there a statistically significant difference between the social and nonsocial conditions.

Reply 13: Thank you for pointing this out. We tested if the two correlations were significantly different and found that they were not ($z=0.754$, $p=0.451$). We report these results on page 35 and also included such a test for the correlation analyses for the EBDM task (which did show a significant difference ($z=-1.96$, $p=0.025$); reported on page 25). We have revised our discussion on this finding to refrain from interpreting null results:

Revision – Manuscript page 31

“Specifically, the difference in subjective feelings of state loneliness between the two isolation sessions (which were two highly controlled active conditions) predicted the difference in reward seeking in the social context of the EBDM task but not in the non-social context. Thus, individuals who reported higher state loneliness also showed stronger social reward seeking following total isolation. We find a similar relationship for the difference in learning rates from negative feedback in the RL task but the difference in correlations between social and non-social feedback was not significant for this task. While these results suggest that, in line with reward seeking, individuals who reported higher state loneliness following isolation also showed stronger reward learning after total isolation, the difference in correlations between social and non-social feedback was not significant for this task. Therefore, we cannot conclude that this finding is specific for reward learning from social feedback (as opposed to reward learning from non-social feedback).”

Comment 14 - fMRI univariate design matrix: According to the methods, “The design matrix included boxcars for the experimental conditions convolved with a double-gamma hemodynamic response function(HRF)”. Could the authors please clarify the duration of these boxcars?

Reply 14: The duration of the boxcars was 2 s. However, we have decided to remove these findings from the present manuscript (see Reply 15 for detailed reasoning).

Comment 15 - fMRI univariate results and ROI analysis: The fMRI results add little to the manuscript in its current form. That said, I wonder more could be reported to further warrant their inclusion. First off, it would be appropriate to provide coordinates (in a table in the supplement) for the reward-sensitive regions identified in the contrast of High Reward > No Reward – as this would help for future meta-analyses and provide a sense of where exactly these regions were (e.g., the occipital cortex is a large region with many brain areas in it – where exactly were the reward-sensitive regions?). Second, it is unclear to me why the Iso+Media condition was completely excluded from the fMRI results. At the very least I would want to see whether there was a relationship between reward sensitivity in the OC and reward-seeking in the Iso+Media condition (i.e., as in figure 8B for the Iso Total and Baseline conditions). The lack of mention of the Iso+Media condition and lack of justification for its exclusion is perplexing. Finally, I wonder why the authors chose to use such a large ROI over the occipital cortex (at least from what I can tell from looking at the inset in Figure 8B)? What parts of the occipital cortex are covered with this ROI? It seems that it may cover retinotopic regions as well as lateral occipital regions. Why not focus the ROI analysis on those regions that were identified as reward sensitive within the ROI (i.e., a combined functional/anatomical ROI). This would ensure targeted and more interpretable results as well as likely boost the statistical power of the results.

Reply 15: Thank you for this comment, which is similar to a comment raised by reviewer 3. After careful consideration and discussion amongst the author team, we have decided to remove the fMRI results from the manuscript. We chose to do this based on the following rationale: We agree with the reviewers that those analyses are currently not well integrated with the main results. The reason for this is that the neural findings address a somewhat separate research question and hypothesis, which concerns the question of whether brain markers predict individual differences in the effects of isolation. This differs from the main research question in this manuscript, which addresses the effects of isolation on reward processing (see preregistration <https://osf.io/kgbgsv> Hypotheses). Furthermore, because the neural findings need to be seen as preliminary, as stated in the manuscript, as the sample size was not properly powered for this type of analysis (i.e., assessing individual differences rather than within-subject changes), we consider these findings to be peripheral findings rather than a central part of the results. In addition, the manuscript is already lengthy, and we think that further expanding on hypotheses and background of the neural findings would not improve the manuscript but make it more diffuse and difficult to follow. Hence, we have decided to remove the neural findings. However, we appreciate the feedback from the reviewer on these data and will incorporate it in any potential future report of these results.

Comment 16 - Discussion of the fMRI results: on page 24 lines 513-514, the authors state: “individual differences in such attention signals to rewards might represent the neural mechanism underlying the contribution of the occipital cortex to the behavioural change.” This assumes that the occipital cortex IS contributing to behavioral change. Correlation with behavioral change is not the same as causally contributing to behavioral change, I would advise rewording.

Reply 16: We have removed this statement from the discussion given our decision to not report these findings in the present manuscript.

Reviewer #2

Comment 1 - The paper presents a compelling investigation into the effects of isolation on reward sensitivity and social feedback processing in adolescents. Overall, the study is well-executed, with clear communication of its rationale, transparent methodology, and appropriate interpretations of the results. Below, I summarize the strengths of the paper and offer suggestions for some minor additions to the discussion section.

Strengths of the Paper

1. Clarity of rationale: The authors provide a clear rationale for their research questions and hypotheses. They draw on existing literature on adolescent social needs, isolation, and reward sensitivity to frame their study, making a persuasive case for examining the effects of short-term isolation on reward responsiveness. I think the theoretical framework is coherent and logically supports their hypotheses on the effect of isolation.
2. Transparency in methodology: The study’s methodology is described in sufficient detail to ensure reproducibility and facilitate replication. The authors clearly outline participant recruitment criteria, experimental procedures, task designs, and analytic strategies. I don’t know computational modelling very well so I couldn’t comment on the specific analyses but I appreciate the details in explaining the models. I also appreciate that the authors also clearly identified where they deviated from the original plan, for example, why they decided to use RT as outcomes due to low variance in choice data.
3. Clear communication of results: The results are presented clearly, with a distinction between findings that directly address the original research questions and those that are exploratory (e.g., correlation between difference in loneliness and difference in RT). The discussion section is thorough in relating results back to the original hypotheses and theoretical framework.
4. Appropriateness of interpretations: The interpretations of results are thoughtful and appropriately cautious, with the authors avoiding unwarranted causal claims and also being very clear about where the results do or do not generalize. They acknowledge that their study is primarily correlational and interpret their findings in light of this limitation.
5. Reflection on limitations: The authors thoughtfully discuss several limitations, including sample size, potential individual differences in baseline social connectivity, and the specificity of the fMRI findings to the occipital cortex. They also suggest meaningful directions for future research, which adds value to the paper.

Reply 1 - We thank the reviewer for this positive evaluation of our work and for their valuable comments on how to further improve the manuscript below.

Comment 2 - Suggestions for Improvement

1. About the justification for why comparing total isolation with isolation with social media, the authors said: “This design allowed us to compare the effects between two tightly controlled and counterbalanced experimental sessions (iso total; iso with media), which differed only in the amount of social interactions participants had, while keeping other factors (such as spending time alone in a room) constant (see methods section for details on procedures)”. I believe the authors could further strengthen this justification by highlighting that this approach is also the most practical. An alternative design involving in-person interactions would require more logistical resources, including additional manpower (e.g., for an RA to act as interaction partner) and careful consideration of the nature of the interaction (e.g., structured interactions like the fast friend paradigm, which has been used in prior

studies as a control condition for solitude). Although, this can be a consideration for future research to consider alternative comparison to total isolation. In contrast, allowing participants to engage in social media provides a logistically feasible and scalable way to operationalize social interaction in a controlled manner, and keep it relatively constant across participant. Am I understanding the choice of iso w media correctly?

Reply 2: We thank the reviewer for this suggestion and have added a statement in the limitations section that future research should consider alternative comparison conditions to social isolation:

Revision – Manuscript page 35

“Our comparison conditions to social isolation were a baseline session and a session where participants were isolated but had access to virtual social interactions. Future studies might explore alternative comparison conditions to the isolation session, such as in-person social interaction in the lab (for example, by providing confederates as interaction partners).”

Comment 3 - Do we see variance in the levels of social media engagement in the iso w media condition? The authors mentioned: “A substantial number of participants (18 out of 40; 45%) reported spending more than 50% of their isolation time participating in virtual social interactions” but do some not interact at all or do it very little?

Reply 3: Thank you for catching this omission. There was indeed variance in how much time participants spent on virtual social interactions. We now report the descriptive statistics for this variable in the results section:

Revision – Manuscript page 22

“(range 5-100%; mean = 47.35, median = 40, std = 28.3)”

We note that this study was not designed to assess social media use in detail and therefore it is not sufficiently powered to assess between-subject differences in social media use and how this relates to effects in this session. However, we agree this would be an important direction for future research and have included a statement on this in the discussion section:

Revision – Manuscript page 34

“In the current study, we asked self-report questions about virtual social interactions during the iso-with-media session and report these in the results section. While our study was not sufficiently powered to assess how between-subject differences in social media use impacted the results for this session, this would be an important direction for future research.”

Comment 4 - While the authors interpret increased reward seeking in the social context as evidence of heightened reward sensitivity following isolation, what come to my mind is whether it is possible that this behavior might also reflect hypervigilance to social cues. There is a link between state loneliness and hypervigilance to social cues. Can the authors elaborate on this distinction in the discussion? Such as to tease apart what evidence in their findings would help us tease apart reward sensitivity versus hypervigilance?

For example, since hypervigilance often entails a stronger response to negative feedback or ambiguous social stimuli, this distinction could be further explored by comparing participants’ responses to positive versus negative feedback; although I couldn’t tell whether that is possible with this data. Such an analysis would help determine whether the observed behavior is better explained by heightened reward sensitivity (i.e., stronger learning from both positive and negative feedback) or hypervigilance (i.e., a stronger bias toward negative social cues).

Reply 4: Thank you for this comment. We did differentiate between responses to positive vs negative feedback in our learning task by modelling two separate learning rates (one from positive prediction

errors (i.e., results that were better than expected, “positive learning rate”) and one from negative prediction errors (i.e., results that were worse than expected, “negative learning rate”).

We found higher positive learning rates for social and non-social feedback following isolation compared to baseline, which is in line with an interpretation of heightened reward responsiveness. We also found lower perseverative errors after total isolation when learning about rewards and faster RTs after isolation during reward learning (regardless of whether feedback was social or non-social). These results suggest that participants showed generally improved reward learning that was not specific to social feedback, indicating that the behaviour is in line with heightened reward responsiveness.

However, we also found that participants showed higher negative learning rates for social feedback (but not non-social feedback) following social isolation. This effect was stronger in participants who reported more loneliness following isolation. This specific finding could also be explained by heightened hypervigilance to negative social cues, and we have revised our discussion to include this possible interpretation:

Revision – Manuscript page 32

“Furthermore, our finding of increased learning from negative social feedback (in addition to increased learning from positive social and non-social feedback) indicates that participants might also show hypervigilance to negative social cues following isolation, which is in line with findings from past research on chronic loneliness^{79,80}.”

Comment 5 - The authors mention arousal briefly as a potential factor influencing reward responsiveness but do not elaborate on its role. Given the link between arousal and heightened sensitivity to environmental stimuli, including social cues, it would be valuable to discuss this more thoroughly in the discussion section.

Reply 5: Thank you for this comment. Because we did not measure arousal it is difficult to say how much our effects might have been driven by this construct. However, we have now added a more in-depth discussion on this issue:

Revision – Manuscript page 31

“Indeed, past research has shown that isolation leads to decreases in general arousal¹¹⁸. However, because low arousal has been associated with reduced motivation in past research⁷⁸, it is unlikely that changes in general arousal were driving our results. Nevertheless, future studies should include measures of arousal and reward responsiveness following isolation to explore any potential interaction between these effects.”

Comment 6 - Overall, I believe this paper makes a significant contribution to the understanding of how social isolation affects reward responsiveness in adolescents. The research is methodologically sound, the rationale is clear, and the results are appropriately interpreted.

Reply 6: We thank the reviewer again for their positive evaluation of our research.

Reviewer #3

Comment 1 - The present manuscript by Tomova and colleagues “Acute isolation is associated with increased reward responsiveness in human adolescents “ investigated how short-term social isolation from personal interaction (with and without access to virtual social interaction) affected self-reported loneliness, reward seeking and reward learning adolescents of 40 young adolescents (16-19 years old). Participants reported to experience increased state loneliness with isolation (in general) as compared to baseline. Access to virtual interaction lead to reduced (state) loneliness reports. In the reward tasks, isolation was associated with faster decisions and “improved” reward learning. The study also employed fMRI and reported an association between lower neural reward sensitivity and “stronger effects of isolation”. The authors conclude mainly that “isolation is associated with higher reward responsiveness”.

Overall this manuscript is addressing a timely and relevant question in psychology regarding effects of experienced loneliness on important aspects of human cognition (here reward/motivation).

Strengths include the use of established behavioral tasks, an experimentally effortful isolation design (consistent with the first author’s prior work), the adolescent age range and the preregistration of analyses.

Weaknesses exist with respect to manuscript structure, (potentially) interpretation of behavioral effects, lack of outlining specific hypotheses in the manuscript, social isolation condition framing, the motivation/justification for integrating neural data and some methodological/statistical aspects that deem further clarification.

Response 1: We thank the reviewer for their positive evaluation of our study and for their valuable comments on how to further improve the manuscript.

Comment 2 - Major comments:

Are the effects on reward task metrics clearly due to the isolation or study design (performing the same task twice). Could the RT effects in reward seeking after isolation for high reward conditions, as well as the RT effects in reward learning be explained by repetition effect (i.e. doing the task twice at baseline and again after isolation)- possibly not due to isolation? The authors may be able to address this (e.g. with prior literature) but it should be clearly stated in the manuscript especially if there is no specific effect between the isolation condition.

Reply 2: We thank the reviewer for this comment. In response to the reviewer’s comment, we have now slightly modified the analysis approach by changing which session is included as the reference session in the mixed effects models (i.e., the session to which the other sessions are compared). We initially had baseline as the reference session, but have changed it to the iso total session, which allows us to test whether effects differ between the two isolation sessions (the two counterbalanced sessions) to rule out repetition effects (in addition to comparing the iso total session to baseline).

We have kept (but streamlined) our sensitivity analyses (reported in the SI) for RTs in reward seeking and reward learning which included session order (i.e., information on the specific order of sessions for each participant) as a predictor in the linear mixed effects model to see whether session order interacted with effects of isolation. We did not find effects of session order or interactions between session and session order for any of our measures. We also kept our sensitivity analysis for the reward learning model where we tested if another model, which was identical to the original winning model (i.e., the fictitious update model) but explicitly included session order as an additional predictor, showed a better fit to the data. This model did not show a better fit, suggesting that session order was not able to explain the variance in the data structure and was not driving the effects.

Our revised analyses now also show a difference in RTs between the two isolation sessions for reward seeking, further suggesting that order effects are not driving these results.

Revision – Manuscript page 23

There was a main effect of session showing lower RTs in the total iso session compared to baseline ($b=0.32$, $t=5.06$, $CI=0.19,0.44$, $P<0.001$) and compared to the iso with media session ($b=0.16$, $t=2.55$, $CI=0.04,0.28$, $P=0.011$; Figure 3A). There was also a session * reward * context interaction showing lower RTs in the iso total compared to iso with media session for high reward trials in the social context (Figure 3B). See Supplementary Table 1 for full results. This indicates that, following total isolation, participants showed increased seeking of high rewards in the social context compared to isolation with access to social media.

Please note that our revised analyses involve log-transforming RTs for both tasks, based on reviewer 1's comment.

Comment 3 - The introduction is very short, general (see next two comments) and lacks specific hypotheses. The manuscript would benefit from more details early on without forcing the reader to go specifically to the preregistration. Upon consultation of the SI and the preregistration document it becomes clear that more additional hypotheses were preregistered. That is not mentioned in the main manuscript and confusing. Please clarify that in the main manuscript briefly. Also, please clarify how/when it was decided which hypotheses were included in the manuscript.

Reply 3: We have expanded the introduction and included our hypotheses from the preregistration. We now also explicitly label which analyses were preregistered and which were exploratory in the main text (methods & results sections) and also briefly state how we deviated from the preregistered analyses in the methods section (which we expand upon in the SI). We noticed that our analyses of self-report measures focused on comparing effects between the start and end of isolation in each isolation session, while our preregistration stated that we would compare effects between sessions (baseline, iso total and iso with media). For the sake of completeness, we added the preregistered analysis in the SI (pages 1-2), but we kept our original analysis in the main text as we think it better captures the effects of isolation on these measures than the analysis we preregistered. We now included a description of this deviation from the preregistration in the SI (*Part 2: Analysis plan*).

Revision – Supplementary Information pages 14-15

We deviated from the preregistered analysis by adding an exploratory analysis testing if self-report measures changed throughout the isolation sessions. To do this, we first compared measures taken at T0 to those taken at T3 using paired sample t-tests. Second, to assess if the change over time (i.e. the slope) was different between the two isolation sessions, we used linear mixed-effects models with the following predictors: session (iso total and iso with media) and duration (hours of isolation: 0, 1, 2, 3) with subject included as a random effect. Note that we did not include the baseline session in this comparison as this session did not have the relevant measures (i.e., repeated collection of state measures throughout isolation).

The command was: `fitlme(Data,'self-report measure~(session*duration)+(session*duration|subjectID)')`.

We included these exploratory analyses in the main text (and the SI on page 2) because we think they better capture the effects of isolation on these measures compared to the analysis we preregistered.

We decided to write a manuscript summarizing the results of the two reward tasks together after data analysis. We initially included the neural results based on the rationale that they were also measuring processes related to reward processing but based on feedback from reviewers 1 & 3 we decided to remove this from the present manuscript. We have included a brief statement explaining how we decided which hypotheses to include in the manuscript in the introduction:

Revision – Manuscript page 4

“This study was part of a larger research project, which included different tasks and tested different hypotheses related to the effects of isolation. The present manuscript focuses on hypotheses related to the effects on reward processing in the domains of reward seeking and reward learning.”

Comment 4 - Is social isolation with access to virtual interaction really “social isolation”? Especially in an age group where virtual social interaction may be an integral part of social interactions? Can the authors specify the respective operationalization of “social isolation” and maybe expand how this may relate to their prior work in adults and speculate on how difference in isolation types may vary in older adults?

Reply 4: We called this session ‘isolation with media’ because participants were objectively isolated from real-life interactions but had access to virtual interactions. Whether that *felt* like social isolation to them or not was one of our research questions. We have added speculation on how different isolation types may vary in older adults in the discussion section:

Revision – Manuscript page 34

“Furthermore, whether these findings are specific to adolescent participants or would also apply to other age groups should be tested in future studies.”

Comment 5 - Motivation for integrating neural data: the introduction is omitting sufficient background, rationalization and hypotheses for investigating brain function within the context of their study motivation. Overall the introductions seems very short and is lacking specific hypotheses.

Reply 5: Thank you for this comment, which is similar to a comment raised by reviewer 1. After careful consideration and discussion amongst the author team, we have decided to remove the fMRI results from the manuscript. We chose to do this based on the following rationale: We agree with the reviewers that those analyses are currently not well integrated with the main results. The reason for this is that the neural findings address a somewhat separate research question and hypothesis, which concerns the question of whether brain markers predict individual differences in the effects of isolation. This differs from the main research question in this manuscript, which addresses the effects of isolation on reward processing (see preregistration <https://osf.io/kbgsv> Hypotheses). Furthermore, because the neural findings need to be seen as preliminary, as stated in the manuscript, as the sample size was not properly powered for this type of analysis (i.e., assessing individual differences rather than within-subject changes), we consider these findings to be peripheral findings rather than a central part of the results. In addition, the manuscript is already lengthy, and we think that further expanding on hypotheses and background of the neural findings would not improve the manuscript but make it more diffuse and difficult to follow. Hence, we have decided to remove the neural findings.

We plan to report these results in a separate format.

Comment 6 -Structure of the article: Consider reversing methods and results more in line with classic article structure, common also in this journal, to help the reader follow the necessary details and adding some less relevant details for understanding results to a SI (instead of a lengthy methods section at the end of the article)? If restructured the results section could be less “methods heavy” and help flow of the overall article. The methodological details in the results may be distracting from the actual results in the current article structure.

Reply 6: We have restructured the manuscript accordingly.

Comment 7 - Did the Correlations between change in self-reported state loneliness and change in RTs in the EBDM task differ statistically between social and non-social feedback (figure 4)? The absence of an effect for non-social context does not imply a differing relationship otherwise (see <https://www.nature.com/articles/nm.2886>). I have the same question/concern for the relationship between difference in state loneliness and difference in learning_neg rates from social feedback btw social and non-social feedback, as well as the relationship between baseline iso total for PSC and RT

The title is implying a focus of the study on the neural data, but that does not reflect the emphasis thereof in the manuscript. Please clarify and adjust if necessary.

Reply 7: We have added statistical tests of difference in correlations for each task. We find that the change in RTs differs significantly between social and non-social context for the EBDM task ($z=-1.96$, $p=0.025$; reported on page 25) but not for the RL task ($z=0.754$, $p=0.451$; reported on page 35).

We have also updated our discussion on these findings:

Revision – Manuscript page 32

“Specifically, the difference in subjective feelings of state loneliness between the two isolation sessions (which were two highly controlled active conditions) predicted the difference in reward seeking in the social context of the EBDM task but not in the non-social context. Thus, individuals who reported higher state loneliness also showed stronger social reward seeking following total isolation. We find a similar relationship for the difference in learning rates from negative feedback in the RL task but the difference in correlations between social and non-social feedback was not significant for this task. While these results suggest that, in line with reward seeking, individuals who reported higher state loneliness following isolation also showed stronger reward learning after total isolation, the difference in correlations between social and non-social feedback was not significant for this task. Therefore, we cannot conclude that this finding is specific for reward learning from social feedback (as opposed to reward learning from non-social feedback).”

Regarding the title: “Reward responsiveness” was meant as an umbrella term including reward seeking (EBDM task) and reward learning (Reward Learning task). We revised the title of the manuscript to improve clarity:

Revision – Title

“Acute isolation is associated with increased reward seeking and reward learning in human adolescents”.

Comment 8 - Can the authors comment in the discussion more on the fact that the only effect on neural reward sensitivity was in the occipital cortex – and not any other – potentially more expected region?

Reply 8: As explained in Reply 5, we have removed the fMRI data from the manuscript.

Comment 9 - Minor comments:

Can the authors briefly specify more details on the non-standard processing of the 7T data or refer to the details in the preregistration? I could not find preprocessing details in the OSF code repository. This is relevant for reproducibility.

The use of 7T in adolescents is relatively novel, if not at least (more) noteworthy. May there be any specific effects/confounds/opportunity for this study in 7T versus the more standard 3T?

Reply 9: Thank you for this comment. As we have removed the fMRI data from this manuscript, we did not include such an addition but will add in more details about non-standard processing of these data in any potential future reports on these data.

Comment 10 - Consider adding a definition of state loneliness earlier in the introduction.

Reply 10: We have added a definition in the introduction:

Revision – Manuscript page 2

“State loneliness refers to the *momentary* feeling of loneliness, which can fluctuate in daily life and is influenced by different social contexts and company^{19,20–22}.”

Comment 11 - Prior research on relationship between isolation and reward seeking is only briefly mentioned (and mainly the authors own work) in the introduction and the discussion.

Reply 11: We have expanded the introduction and discussion including more prior research on this topic.

Introduction:

Revision – Manuscript pages 2-3

“Here we were interested in studying how changes in state loneliness induced via social isolation impact subsequent processing of rewards in adolescents. The domain of reward is of particular interest due to its powerful effects on behaviour²⁴. There have been persistent efforts to identify the impacts of social isolation on reward processing in adolescents using animal models of isolation. These studies have consistently shown that isolating adolescent animals primarily affects dopaminergic brain reward circuits and alters reward seeking and reward learning^{25,26}.

For example, social isolation in adolescence has been shown to increase sensitivity to social rewards²⁷ and also increase seeking of food or drug rewards in rodents²⁸. Conversely, operant access to social interaction has been shown to prevent drug self-administration in rats²⁹. Animal research has further shown that social isolation also changes learning about rewards in adolescent animals. Specifically, studies have found that adolescent isolation leads to improved learning about stimulus-reward associations^{30,31} but causes perseveration to initially learned associations when reward contingencies change³²⁻³⁴. Thus, animal research suggests that while isolation can improve initial learning about rewards, it diminishes the flexibility of changing initially learned responses.

While these results from animal studies have potential implications for the effects of social isolation on reward processing in humans, it is unclear to what extent these findings can be translated to human adolescents.”

Discussion:

Revision – Manuscript page 30

“Our findings are partially consistent with results from animal models, which have shown that social isolation in adolescent rodents increases reward responsiveness²⁶. In animals, social isolation increases responsiveness to social rewards⁷⁴, seeking of food or drug rewards and risk of developing addictions²⁸. Animal models have also shown that isolation improves learning of stimulus-reward associations^{30,31} (but not their reversal³²⁻³⁴). Our findings suggest that there is a similar effect of increased responsiveness to rewards immediately following total isolation in human adolescents. These findings are also in line with human research on chronic loneliness (in adults) showing that individuals high in loneliness show stronger responsiveness to social rewards⁷⁵⁻⁷⁷.”

Revision – Manuscript pages 33-34

Chronic and state loneliness can have opposite effects on social behaviour⁸³. The social homeostasis model suggests that prolonged engagement of mechanisms intended for short-term adaptation to state loneliness can cause pathological states during chronic loneliness⁸⁴. This model was proposed based on research on the effects of isolation in animal models, but similar theoretical frameworks have been proposed based on human research on chronic loneliness⁸⁵. For example, the cognitive control model of loneliness⁸⁵ similarly proposes that it is the prolonged engagement of regulatory mechanisms during loneliness that exhausts cognitive resources and results in aversive outcomes.

Comment 12 - 10 social or virtual interactions seems low (at least without further reference). The authors say “frequent” but is there normative data that can be related to possibly? Similar comment for the chronic loneliness levels.

Reply 12: As we describe in the methods section (*Screening questionnaire; page 5*), we compared these numbers to averages collected in previous studies with the same population (adolescents) and included references to the relevant studies:

“As we aimed to study the effects of isolation in a sample of adolescents who have frequent and regular social interactions, we also excluded people who: (1) lived alone; (2) reported high feelings of chronic loneliness on the UCLA loneliness scale²³ (we excluded adolescents with scores >50, which is 2 standard deviations above the mean for an adolescent sample⁴²); and/or (3) reported substantially smaller social network sizes than previously reported for an adolescent sample⁴³ measured via two measures: (a) Number of close friends (the original questionnaire asks for the number of people who give social support⁴⁴, which we adapted to the number of close friends to simplify the question for our adolescent sample); and (b) Number of social interactions in the past month: counting face-to-face and virtual social interactions that were primarily social (i.e., excluding professional interactions, like talking to a doctor, teacher, or hairdresser). We excluded participants who reported fewer than two close friends and fewer than 10 social interactions in one month (which is ~7 standard deviations below the previously reported means for adolescents for both measures⁴³).”

Comment 13 - In the beginning of the discussion (pages 20/21/22) behavioral findings on reward behavior is stated as being consistent with findings in animals – but the link to human work should/could be highlighted more.

Reply 13: We thank the reviewer for this comment. As we state in the introduction, the majority of work on the effects of isolation on reward processing has been done in animals so far. Human work so far has focused on correlational designs studying the association between chronic loneliness and reward behaviour. To the best of our knowledge, this is the first study to use an experimental approach to study the effects of isolation on reward processing in human adolescents.

In response to the reviewer’s comment, we have included a statement in the discussion to include studies which have assessed the link between chronic loneliness and reward processing:

Revision – Manuscript page 30

“These findings are also in line with human research on chronic loneliness (in adults) showing that individuals high in loneliness show stronger responsiveness to social rewards ^{75–77} .”

Comment 14 - Methodological details that could/should be elaborated on:

- Specify fMRIPrep version.
- On pages 49/50, information on framewise motion exclusion criteria seems to contradict? (ART vs FMRIPREP, FD vs FD&spike intensity?) Please clarify.
- What were the MRIQ reports used for? Was any data excluded based on IQMs?

Reply 14: We have removed these data from the current version of the manuscript but will add all this information to any potential future reports of these data.

Comment 15 - Please specify/approximate effect sizes for interactions.

Reply 15: For each main effect and interaction, we are reporting unstandardized estimates as effect sizes which is in line with recommendations on how to report effect sizes (e.g., Pek & Flora, 2018). There is no clear way to calculate standardized effect sizes for individual model terms such as main effects or interactions for linear mixed-effects models due to the way how variance is partitioned in these models (e.g., Rights & Sterba, 2019). However, mixed models have the advantage of being superior in controlling for Type I errors and non-independence than alternative approaches which is why we chose to implement them (e.g., Judd et al. 2012; Barr et al. 2013).

References

Barr, D. J., Levy, R., Scheepers, C., & Tily, H. J. (2013). Random effects structure for confirmatory hypothesis testing: Keep it maximal. *Journal of Memory and Language*, 68(3), 255–278. <https://doi.org/10.1016/j.jml.2012.11.001>

Judd, C. M., Westfall, J., & Kenny, D. A. (2012). Treating stimuli as a random factor in social psychology: A new and comprehensive solution to a pervasive but largely ignored problem. *Journal of Personality and Social Psychology*, 103(1), 54–69. <https://doi.org/10.1037/a0028347>

Pek, J., & Flora, D. B. (2018). Reporting effect sizes in original psychological research: A discussion and tutorial. *Psychological Methods*, 23, 208–225. <https://doi.org/10.1037/met0000126>

Rights, J. D., & Sterba, S. K. (2019). Quantifying explained variance in multilevel models: An integrative framework for defining R-squared measures. *Psychological methods*, 24(3), 309. <https://doi.org/10.1037/met0000184>

Comment 16 - In the SI, please provide full statistical details (not just “all p values > 0.x”)

Reply 16: Thank you for this comment. We have added full statistical details in the SI.

Comment 17 - In SI – MRI data analysis: Instead of quoting sections from the preregistration that were executed as preregistered, consider describing what was done instead? Otherwise it is hard to decipher and potentially redundant to the methods section.

Reply 17: As we have removed the fMRI results from the manuscript, we have also removed this section from the SI.

Reviewer #1

The authors did an excellent job of addressing the majority of my concerns and the paper reads very well – I have one minor comment but besides that I’m happy to recommend this for publication.

Minor Comment:

Comment 1 - Page 13: L 403-405: “However, we were not able to run this analysis due to the low variance in participants' choice data. More specifically, many participants chose to play almost every trial in the task (73% of all trials were played).” 73% of trials seems to leave quite a bit of room for variance, perhaps the authors could include the between participant standard deviation for this metric? Alternatively, it would be helpful to know how many participants played 100% of trials.

Reply 1: We thank the reviewer for their positive evaluation of our revision. We added the additional information in the methods section.

Revision – Manuscript page 13

“More specifically, many participants chose to play almost every trial in the task (73% of all trials were played; **between participant standard deviation = 20.2%; 15 participants played 100% of trials in at least one of the sessions**).

Reviewer #4

I was invited to comment on the revised manuscript without having reviewed the initial version of the paper. Specifically, I was asked to evaluate the changes made in response to Reviewer 3’s comments.

Overall, I concur with the other reviewers that the manuscript addresses a timely, relevant, important, and scientifically valid research question on the impact of short-term social isolation on affect and reward processing in adolescents. I also share their enthusiasm regarding the quality of the research and its execution, the amount of work and effort that was dedicated to producing the manuscript, the clear description of the method, the calibrated interpretations, and the adoption of rigorous open-science practices promoting transparency. I particularly appreciated the inclusion of the supplementary note detailing the deviations from the preregistration in a clear and transparent manner, which is an excellent practice.

With respect to the revised manuscript, the authors have provided a clear and detailed revision that satisfactorily addressed most of the Reviewer 3’s comments. The updated analytical approach (i.e., using the iso total session as reference rather than the baseline session) and the control analyses including session order converged in suggesting that the RT effects in the reward tasks are unlikely to be driven by practice/repetition effects, but more likely rely on the social isolation manipulation. The introduction now integrates well previous animal and human research on the effects of social isolation, clarifies how social isolation is defined, conceptualised, and operationalised, and clearly specifies the hypotheses.

These improvements notwithstanding, I believe there is still room to clarify and better streamline the presentation of the results. As it stands, the results section is somewhat difficult to follow. It is occasionally unclear which results correspond to which hypothesis, why some hypothesis-related results are reported in the supplementary information rather than in the main text, and why some results are presented in figures without being described in the text—despite being referenced in the discussion. These issues make it challenging to connect the discussion of the findings with the statistical results. It would also be important to clarify the extent to which the comparison between the two social isolation conditions (i.e., iso total vs. iso with media) is central to the hypotheses. Additionally, it would be beneficial to report in the main text how the sample size was determined, along with a justification. Some analytical decisions would also deserve to be more thoroughly

justified. I elaborate on these points, along with other minor comments, in detail below. I hope these suggestions will be constructive and helpful in strengthening the manuscript even further.

We thank the reviewer for their positive evaluation of our work and for the constructive feedback on how to further improve the clarity of the manuscript.

Primary comments

Comment 1 - The organisation of results section could be improved to make it easier for the reader to connect the statistical results with their interpretation in the discussion. As currently written, the results section is somewhat challenging to follow and integrate. It would be particularly helpful to clearly link the relevant results to the specific hypothesis being tested. This would better streamline the presentation of findings and clarify the extent to which they provide a compelling answer to the hypotheses.

Reply 1: Thank you for this constructive suggestion. We have included statements that link the relevant results to the specific hypothesis being tested throughout the results section:

Revision – Manuscript page 22

“Our analysis testing effects of isolation on self-report measures of affect (H1), showed that after three hours in both isolation sessions, compared with at the start of the session, participants reported significantly increased state loneliness, boredom and decreased positive mood (Table 1; Figure 2).”

Revision – Manuscript page 22

“Our analysis testing whether social media use during isolation would remediate effects of isolation on self-report measures of affect (H4), showed that the increase in state loneliness and boredom over time was higher in the iso total session than in the iso with media session (state loneliness: beta (b)=-3.76, t(302)=-2.89, 95% confidence interval (CI)=-6.33, -1.20, P=0.004; boredom: b=-6.01, t(302)=-3.43, CI=-9.45, -2.57, P<0.001).”

Revision – Manuscript page 23

“Our analysis testing effects of isolation on reward seeking (H2) and whether social media use during isolation would remediate effects of isolation on reward seeking (H4), showed that there was a main effect of session showing lower RTs in the iso total session compared to baseline (b=0.32, t(732)=5.06, CI=0.19,0.44, P<0.001) and compared to the iso with media session (b=0.16, t(732)=2.55, CI=0.04,0.28, P=0.011; Figure 3A).”

Revision – Manuscript page 25

“To test the effects of isolation on reward learning (H3) and whether social media use during isolation would remediate effects of isolation on reward learning (H4), we extracted the population-level mean posterior distributions of each parameter estimated by the model: learning rates (learning_pos and learning_neg) and exploration (beta).”

Comment 2 - In a similar vein, I think it would be important to report all primary analyses testing the hypotheses under consideration in the main text rather than in the supplementary information. This is particularly relevant for the hypothesis concerning the effects of social isolation on self-reported affect (i.e., increased loneliness, craving for social contact, state anxiety and negative mood, and decreased positive mood; H1), for which only the results on state loneliness and positive mood are currently included in the main text. My suggestion would be to (a) report the statistical results for all self-report dependent variables in a table, (b) briefly summarise the results in the main text, and (c) incorporate Supplementary Figure 3 into Figure 2. These changes would help present the results for H1 in a more comprehensive, concise, and accessible manner.

Reply 2: Thank you for this comment. We now (a) report the statistical results for self-report dependent variables in a table:

Revision – Manuscript page 22

Table 1 Results pairwise comparisons T0 vs T3 during isolation for self-report measures of affect for both isolation sessions (iso total and iso with media; N = 40).

Measure (T0 vs T3)	t (df)		p		Effect size (Cohen's d)	
	Iso total	Iso w media	Iso total	Iso w media	Iso total	Iso w media
State Loneliness	6.79 (39)	4.50 (39)	<0.001	<0.001	1.04	0.63
Positive Mood	-6.23 (39)	-6.23 (39)	<0.001	<0.001	0.74	0.74
Negative Mood	-0.32 (38)	-0.68 (38)	0.753	0.503	0.06	0.13
Boredom	8.32 (38)	4.17 (39)	<0.001	<0.001	1.45	0.65
Social Craving	-0.94 (39)	-0.38 (38)	0.353	0.709	0.13	0.06

b) briefly summarise the results in the main text:

Revision – Manuscript page 22

“Our analysis testing effects of isolation on self-report measures of affect (H1), showed that after three hours in both isolation sessions, compared with at the start of the session, participants reported significantly increased state loneliness, boredom and decreased positive mood (Table 1; Figure 2).”

(c) and incorporated Supplementary Figure 3 into Figure 2:

Revision – Figure 2

FIGURE 2 Effects of isolation on self-report measures of affect.

Changes in self-reported state loneliness (A), positive mood (B), negative mood (C), social craving (D), boredom (E) over time during isolation. State anxiety (F) measures for each session. Dark green = iso

total session; turquoise = iso with media session. The boxplots indicate the median (black centre line), mean (white circle), the interquartile range (IQR; box) and the 1.5 x IQR minima and maxima (whiskers). The black rhombi indicate outliers (values outside of 1.5 x IQR). The dashed lines across the plots indicate the mean (white) and median (grey) ratings during baseline and their 95% confidence interval (grey area). * indicates $p < 0.001$.

Please note that state anxiety results are not in Table 1 as it was measured only once during isolation, preventing assessment of changes over time. We report between-session comparisons, along with other self-report affect measures, in the supplement, as noted in the main text:

Revision – Manuscript page 23

“State anxiety did not differ between sessions (Figure 2 and see Supplementary Note 2 for details). We report the direct comparison between sessions for the other measures (comparing measures after 3 hours of isolation) in the Supplementary Note 2.”

Comment 3 - The results comparing the two social isolation conditions (i.e., iso total and iso with media) were not always consistently reported in the text, notably regarding the estimated parameters extracted from the computational models of the reward-learning task. It would be extremely helpful to clearly report these differences—even when they are not statistically significant—whenever they are relevant to the hypotheses, especially if they are discussed later in the manuscript. Doing so would help the reader better understand the connection between the modelling results (e.g., p. 32, ll. 918-931) and their interpretation.

Reply 3: If we understand the comment from the reviewer correctly, this information is already reported in Figure 6B, which depicts the difference scores between iso total and iso with media for each parameter extracted from the computational model of the reward learning task (including a Bayes Factor quantifying the support for the alternative hypothesis for each pairwise comparison). In response to the reviewer’s comment, in the discussion we now indicate where the relevant information in the results section can be found. We hope this is now clearer.

Revision – Manuscript page 30

“They also showed no difference in reward learning from positive prediction errors (in the RL task), compared with baseline (**Figure 6B, first row, column: Difference: iso w media - baseline**) and more perseveration compared to the isolation session (while there was no difference in perseveration compared to baseline; **Figure 5D**). However, learning rates from negative prediction errors for social feedback were higher in both isolation sessions compared to baseline (**Figure 6B, second row, columns: Difference: iso w media – baseline / Difference: iso total - baseline**).”

Comment 4 - Relatedly, it would be important to clarify the importance of the comparisons between the social isolation conditions in relation to the manuscript’s overall contribution. Because these comparisons are not always consistently reported or discussed, it remains somewhat unclear whether they are central to the hypotheses. Expanding on this aspect would enhance the clarity of the manuscript and help the reader better situate and interpret its contributions.

Reply 4: We thank the reviewer for highlighting the lack of clarity regarding this. The comparisons between the two isolation conditions relate to hypothesis 4, where we predicted that access to social media remediates effects of isolation. In response to the reviewer’s comment (and to comment 1 above), we have added statements connecting the reported results to this hypothesis where relevant (see reply 1 for details).

Comment 5 - Although the preregistration document clearly explains how the sample size was determined via an a priori power analysis based on pilot data, it would be highly beneficial to report this power analysis in the main text. Doing so would provide a transparent justification for the sample size (see Lakens, 2022) and make this key information readily available to the reader without requiring them to consult the preregistration. For completeness, it would also be warranted to specify

the statistical test used in the power analysis to compute the required sample size (e.g., paired-sample t test).

Reply 5: We thank the reviewer for this helpful suggestion. We have added this information to the methods section.

Revision – Manuscript page 4

“As preregistered (<https://osf.io/kbgsv>), we determined sample size based on pilot data (N = 19; taken from part of the sample in Tomova et al. 2020) from 18–24-year-olds which showed that short-term isolation affected feelings of loneliness (using a self-report loneliness scale ranging from 0-100) after just four hours of isolation with an effect size (Cohen’s d) of 0.47. A power analysis (paired t-test) showed that 38 participants are required to detect a medium effect size of $d = 0.47$ in our outcome measures to achieve a power of .80 (1-beta) at an alpha of .05.”

Comment 6 - Regarding the effects of social isolation on reward-seeking (see p. 23, ll. 685-689), I think it would be necessary to conduct a follow-up test to substantiate the interpretation that adolescents increased seeking of high rewards in the social context more after total isolation than after isolation with access to social media. While the three-way interaction between session, reward, and context suggests that the effect of reward level on reward-seeking depends on the social context—and that this interaction itself depends on the session—it does not clearly indicate which specific levels of the independent variables are driving the effect. A pairwise comparison directly testing RTs in the high-reward condition within the social context, contrasting total isolation with isolation with media, thus appears warranted to provide more direct evidence for the interpretation offered.

Reply 6: We have added this information.

Revision – Manuscript page 23

“To help interpret the 3-way interaction, we performed a pairwise comparison directly testing RTs in the high reward condition within the social context, between the iso total and the iso with media session. This showed that RTs in the high reward condition in the social context were significantly lower in the iso total session ($t(39)=2.68$, $P = 0.009$) compared to the iso with media session.”

Secondary comments

Comment 7 - On page 3 (ll. 98-99), the hypothesis (H4) concerning the modulatory effects of the social isolation with media includes only reward-seeking and reward learning. However, in the preregistration, it seems that this hypothesis (H7) also includes self-report measures of affect. If this is correct, it would be worth explaining or justifying this apparent discrepancy.

Reply 7: Thank you for catching this omission. We have now included hypothesis 7 in the manuscript by incorporating self-report measures of affect to hypothesis 4.

Revision – Manuscript page 3

“We hypothesised that (4) social media use during isolation would remediate the effects of isolation on self-report measures of affect, reward seeking and reward learning.”

Comment 8 - On page 4 in the participants section (l. 120), it would be appropriate to report the number of both adolescent women and men, as well as how sex/gender was determined, in order to align with the editorial policies of Communications Psychology.

Reply 8: We have added the requested information.

Revision – Manuscript page 4

“We collected data from 42 participants; 2 participants were unable to complete all experimental sessions, leaving 40 complete datasets (mean age = 17.1; std = 0.9; 22 female, **18 male, gender determined via self-report**).”

Comment 9 - Regarding the analysis of the self-report measures (see p. 12, ll. 356-360), it was unclear to me why only the measures taken at T0 and T3 (or T4, as reported in the supplementary information) were compared rather than including all timepoints in a single statistical model for each self-report measure. I think that a brief justification for these analytical specifications would be particularly helpful.

Reply 9: We chose to compare t0 vs t3 (and t4 in the SI) as we wanted to assess whether loneliness (and other self-report measures of affect) were significantly different at the end of isolation (compared to the beginning), just before participants underwent the reward tasks. However, as described in the methods section, we also ran an analysis including all timepoints in one statistical model in which we tested whether effects differ between the two isolation sessions (iso total vs iso with media). We initially did this only for loneliness and positive mood (reported in the main text), but we have now run this analysis for all self-report measures, in line with moving them all to the main text in response to reviewer comment 2.

Revision – Manuscript pages 22-23

“Our analysis testing whether social media use during isolation would remediate effects of isolation on self-report measures of affect (H4), showed that the increase in state loneliness and boredom over time was higher in the iso total session than in the iso with media session (state loneliness: beta (b)=-3.76, t(302)=-2.89, 95% confidence interval (CI)=-6.33, -1.20, P=0.004; boredom: b=-6.01, t(302)=-3.43, CI=-9.45, -2.57, P<0.001). However, the decrease in positive mood over time was not significantly different between the iso with media and iso total session (b=-0.05, t(302)=-0.18, CI=-0.67, 0.57, P=0.857). There was no difference between sessions in measures over time for negative mood (b=0.003, t(302)=0.02, CI=-0.25, 0.26, P=0.985) and social craving (b=1.22, t(302)=0.92, CI=-1.38, 3.81, P=0.358). State anxiety did not differ between sessions (Figure 2 and see Supplementary Note 2 for details).”

We have also added a justification for focusing on T0 vs T3 (and T4) in the methods section.

Revision – Manuscript page 12

“We chose to compare measures between T0 and T3 (and T4 in the Supplementary Note 1) as we wanted to assess whether loneliness (and other self-report measures of affect) were significantly different at the end of isolation (compared to the beginning), just before participants underwent the reward tasks.”

Comment 10 - On page 14 (ll. 425-430) and page 21 (ll. 631-635), it may be worth specifying whether the variables used in the Pearson correlations were approximately normally distributed. If this was not the case, it would be advisable to additionally or alternatively report Spearman correlations.

Reply 10: Thank you for this comment. The variables included in the correlations were normally distributed as tested via Shapiro-Wilk tests. We have included a statement on this in the relevant sections.

Revision – Manuscript pages 14/21

“The variables used for the correlations were normally distributed as tested using Shapiro-Wilk tests.”

Comment 11 - In the results section and in the supplementary information, it would be beneficial to report the degrees of freedom associated with the t-statistics extracted from the linear mixed-effects models.

Reply 11: Thank you for this suggestion, we now report the degrees of freedom associated with the t-statistics from the linear mixed-effects models in the results section and the SI.

Comment 12 - On page 25 (ll. 731-732), I would recommend avoiding characterizing the lower accuracy in phase 3 during the reward-learning task as a “trend”. Instead, I suggest interpreting all p-values above the alpha level as not statistically significant. The lower accuracy in phase 3 could be described as descriptively aligning with the pattern observed in phases 2 and 4, with the added caveat that this difference did not reach statistical significance.

Reply 12: Thank you for catching this. We have reworded the section accordingly.

Revision – Manuscript pages 24

“Accuracy was numerically, but not significantly, lower in phase 3, when reward contingencies were the same as during acquisition ($b=-0.10$, $t(936)=-1.90$, $CI=-0.20,0.003$, $P=0.057$).”

Comment 13 - On page 27 (ll. 774-774), it would be worthwhile to cite Palminteri and Lebreton (2022) with respect to the asymmetry between learning rates for positive versus negative prediction errors. This paper presents robust evidence for such asymmetry across different contexts and species.

Reply 13: We have added the reference.

Comment 14 - On page 32 (ll. 915-916), the interpretation that the “findings provide support for the potential of social media to fulfil social needs” could benefit from further nuance. Specifically, it would be helpful to acknowledge the fact that social isolation occurred for only relatively short periods of time, which could influence the effects of social media on fulfilling social needs. It remains possible that the effects of social media may differ in the context of longer or prolonged periods of social isolation. Including this caveat would contribute to a more calibrated and context-sensitive interpretation of the potential effects of social media during social isolation.

Reply 14: Thank you for this important comment. We have included a caveat to our statement as suggested by the reviewer.

Revision – Manuscript page 30

“However, we note that in the present study social isolation occurred only for a relatively short period of time, which could influence the effects of social media on fulfilling social needs. It is possible that the effects of social media may differ in the context of longer or prolonged periods of social isolation.”

Signed,
Yoann Stussi

References

- Lakens, D. (2022). Sample size justification. *Collabra: Psychology*, 8(1), Article 33267. <https://doi.org/10.1525/collabra.33267>
- Palminteri, S., & Lebreton, M. (2022). The computational roots of positivity and confirmation in reinforcement learning. *Trends in Cognitive Sciences*, 26(7), 607-621. <https://doi.org/10.1016/j.tics.2022.04.005>